# How Out-of-Distribution Detection Learning Theory Enhances Transformer: Learnability and Reliability

Yijin Zhou [1 2]  Yutang Ge [1]  Wenyuan Xie [1]  Linqian Zeng [1]  Xiaowen Dong [3]  Yuguang Wang [1]

## Abstract

Transformers excel in natural language processing and computer vision tasks. However, they still face challenges in generalizing to Out-of-Distribution (OOD) datasets, *i.e.* data whose distribution differs from that seen during training. OOD detection aims to distinguish outliers while preserving in-distribution (ID) data performance. This paper introduces the OOD detection Probably Approximately Correct (PAC) Theory for transformers, which establishes the conditions for data distribution and model configurations for the OOD detection learnability of transformers. It shows that outliers can be accurately represented and distinguished with sufficient data under certain conditions. The theoretical implications highlight the trade-off between theoretical principles and practical training paradigms. By examining this trade-off, we naturally derived the rationale for leveraging auxiliary outliers to enhance OOD detection. Our theory suggests that by penalizing the misclassification of outliers within the loss function and strategically generating soft synthetic outliers, one can robustly bolster the reliability of transformer networks. This approach yields a novel algorithm that ensures learnability and refines the decision boundaries between inliers and outliers. In practice, the algorithm consistently achieves state-of-the-art (SOTA) performance across various data formats. Code is released under this URL.

## 1. Introduction

Mainstream machine learning algorithms typically assume data independence, called in-distribution (ID) data

(Krizhevsky et al., 2012; He et al., 2015). However, in practical applications, data often follows the "open world" assumption (Drummond & Shearer, 2006), where outliers with different distributions can occur during inference. This real-world challenge frequently degrades the performance of AI models in prediction tasks. One remedy is to incorporate OOD detection techniques. OOD detection aims to identify and manage semantically distinct outliers, referred to as *OOD data*. It requires the designed algorithm to detect and avoid making predictions on OOD instances, while maintaining robust performance on ID data.

The transformer, a deep neural network architecture that leverages attention mechanism, is renowned for its exceptional capabilities in various deep learning models (Vaswani, 2017). It is utilized as a backbone network for OOD detection (Koner et al., 2021; Graham et al., 2022; Hendrycks et al., 2020). Despite the significant performance improvements, the design of OOD detection strategies largely relies on empirical intuition, heuristics, and experimental trial-and-error. There is a lack of theoretical understanding regarding the properties and limitations of transformers for OOD detection, with formal analysis of their reliability being notably scarce. Given that OOD detection is critical to the safety and reliability of deep learning models, there is an urgent need to establish robust theoretical principles in the domain. To foster an intuitive understanding for a broad audience before introducing formal theoretical results, we first present qualitative explanations of **learnability** and **Jackson-type bounds**. Learnability means a model can grasp true patterns from training samples, ensuring its performance increases on unseen data as more samples are provided. Jackson-type approximations provide quantitative upper bounds on the approximation error of neural networks or polynomials in terms of the regularity of the target function (e.g., Sobolev or Lipschitz smoothness), revealing how model parameters like depth, width, and attention configurations influence the approximation rates (Jackson, 1930; Jiang & Li, 2024).

Subsequently, we will provide a rigorous definition of learnability tailored for OOD detection tasks. As an impressive work on OOD detection theory, Fang et al. (2022) defines strong learnability for OOD detection and has applied its PAC learning theory to FCNN-based hypothesis spaces,

---

[1]Shanghai Jiao Tong University, China [2]Shanghai Innovation Institute, China [3]University of Oxford, UK. Correspondence to: Yuguang Wang <yuguang.wang@sjtu.edu.cn>.

*Proceedings of the 43$^{rd}$ International Conference on Machine Learning*, Seoul, South Korea. PMLR 306, 2026. Copyright 2026 by the author(s).

which consist of OOD detectors built upon fully connected neural networks (FCNNs), and to score-based hypothesis spaces, encompassing algorithms that perform OOD detection by employing a score-based strategy subsequent to an FCNN stage.

**Definition 1.1** (Fang et al. (2022), Strong learnability)**.** OOD detection is strongly learnable in $\mathcal{D}_{XY}$, if there exists an algorithm $\mathbf{A}: \cup_{n=1}^{+\infty} (\mathcal{X} \times \mathcal{Y})^n \to \mathcal{H}$ and a monotonically decreasing sequence $\epsilon(n)$ *s.t.* $\epsilon(n) \to 0$, as $n \to +\infty$, and for any domain $D_{XY} \in \mathcal{D}_{XY}$,

$$\mathbb{E}_{S \sim D_{X_I Y_I}^n} [\mathcal{L}_D^\alpha(\mathbf{A}(S)) - \inf_{h \in \mathcal{H}} \mathcal{L}_D^\alpha(h)] \le \epsilon(n), \forall \alpha \in [0,1]. \quad (1)$$

$\mathcal{X}$ and $\mathcal{Y} := \{1, 2, \cdots, K, K+1\}$ denote the whole dataset and label space, $\mathcal{D}_{XY}$ is data domain, $D_{X_I Y_I}^n \subset D_{XY}$ is ID training data with amount $n$.

**Theorem 1.2** (Fang et al. (2022), Informal, learnability in FCNN-based and score-based hypothesis spaces)**.** *If $l(y_2, y_1) \le l(K+1, y_1)$ for any in-distribution labels $y_1$ and $y_2 \in \mathcal{Y}$, and the hypothesis space $\mathcal{H}$ is FCNN-based or corresponding score-based, then OOD detection is learnable in the separate space $\mathcal{D}_{XY}^s$ for $\mathcal{H}$ if and only if $|\mathcal{X}| < +\infty$.*

Inspired by Theorem 1.2, the goal of our theory is to answer the following questions:

*Given a transformer hypothesis space, what are necessary and sufficient conditions to ensure the learnability of OOD detection? Additionally, we aim to derive the approximation rates and error bounds for OOD detection, providing a rigorous theoretical foundation for understanding its performance and limitations.*

We introduce a theoretical framework to analyze the conditions and error boundaries for OOD detection in transformers. Theorem 3.2 shows that penalizing the misclassification of OOD in training loss clarifies the decision boundary between inliers and outliers. This condition ensures that the model achieves *OOD Detection Learnability*, enabling it to reliably distinguish between ID and OOD data. Moreover, we quantify the learnability by proving an error bound linked to the model's depth and budget, specifically the number of trainable parameters (Theorems 3.4 and 3.5).

Due to the complexity of real-world data and the tendency of models to converge to local optima rather than global optima during training, validating these theoretical findings through numerical experiments poses a big challenge. Nevertheless, the theoretical results provide a robust foundation for applying transformers to OOD detection tasks. Additionally, we offer a fresh perspective on prerequisites for achieving learnability, and show the benefits of incorporating ID vs. OOD classification penalties and leveraging auxiliary OOD data during training to improve reliability.

Based on the theory, we propose a new algorithm for transformer networks, named **G**enerate **R**ounded **O**OD **D**ata (GROD), designed to fine-tune transformer networks and improve their ability to predict unknown distributions. By incorporating OOD Detection into the network training process, we can strengthen the recognition of ID-OOD boundaries. When the network depth is sufficiently large, the GROD-enhanced transformer converges to the target mapping, exhibiting robust reliability.

In summary, our **main contributions** are as follows: **(1)** We establish a PAC learning framework for OOD detection applied to transformers, providing necessary and sufficient conditions for learnability, regarding dataset distribution, training strategy and transformer capacity. **(2)** Further, if the transformer capacity is limited to achieve learnability, we prove the approximation rates and error bound estimates for OOD detection regarding model capacity. Theoretical contributions support practical decisions regarding model and training strategy design of learnability and reliability. **(3)** We propose a novel OOD detection approach GROD. This strategy is theoretically grounded and high-quality in generating and representing features regardless of data types, displaying SOTA performance on image and text tasks.

## 2. Notations and Preliminaries

**Notations.** We begin by summarizing notations about OOD detection learnability and transformer architectures. Firstly, $|\cdot|$ indicates the count of elements in a set, and $\|\cdot\|_2$ represents the $L_2$ norm in Euclidean space. Formally, $\mathcal{X}$ and $\mathcal{Y} := \{1, 2, \cdots, K, K+1\}$ denote the whole dataset and its label space. As subsets in $\mathcal{X}$, $\mathcal{X}_{\text{train}}$, $\mathcal{X}_{\text{test}}$, $\mathcal{X}_I$ and $\mathcal{X}_O$ represent the training dataset, test dataset, ID dataset, and outliers respectively. $\mathcal{Y}_I := \{1, \cdots, K\}$ denotes the ID label space, $\mathcal{Y}_O := \{K+1\}$. $l(\mathbf{y}_1, \mathbf{y}_2), \mathbf{y}_1, \mathbf{y}_2 \in \mathcal{Y}$ denotes the paired loss of the prediction and label of one data, and $\mathcal{L}$ denotes the total loss. The data domain priori-unknown distribution space $\mathcal{D}_{XY}$ *i.e.* $\forall D_{XY} \in \mathcal{D}_{XY}$, $\alpha \in [0,1)$, $((1-\alpha)D_{X_I Y_I} + \alpha D_{X_O Y_O}) \in \mathcal{D}_{XY}$, such as $\mathcal{D}_{XY}^{all}$, which is the total space including all distributions; $\mathcal{D}_{XY}^s$, the separate space with distributions that have no ID-OOD overlap; $\mathcal{D}_{XY}^{D_{XY}}$, a single-distribution space for a specific dataset distribution denoted as $D_{XY}$; $\mathcal{D}_{XY}^F$, the finite-ID-distribution space containing distributions with a finite number of ID examples; and $\mathcal{D}_{XY}^{\mu,b}$, the density-based space characterized by distributions expressed through density functions. A superscript may be added in $D_{XY}$ to denote the number of data points in the distribution. The model hypothesis space is represented by $\mathcal{H}$, and the binary ID-OOD classifier is defined as $\Phi$. These notations, consistent with those used in Fang et al. (2022), facilitate a clear understanding of OOD detection learning theory.

Several notations related to spaces and measures of function

approximation also require further clarification to enhance understanding of the theoretical framework. $\mathcal{C}$ and $C$ denote the compact function set and compact data set, respectively. Measures $C_0(\cdot)$, $C_1^{(\alpha)}(\cdot)$ and $C_2^{(\beta)}(\cdot)$ indicate the approximation capabilities of transformers, with $\alpha$ and $\beta$ being the convergence orders for Jackson-type estimation regarding self-attention blocks and feed-forward neural networks. $\widetilde{\mathcal{C}}^{(\alpha,\beta)}$ within $\mathcal{C}$ is the function space where Jackson-type estimation is applicable. Given the complexity of the mathematical definitions and symbols involved, we aim to provide clear explanations to facilitate a smooth understanding of our theoretical approach. These mathematical definitions of the approximation of functions follow Jiang & Li (2024).

**The transformer hypothesis space.** Under the goal of investigating the OOD detection learning theory on transformers, our research defines a fixed transformer hypothesis space for OOD detection $\mathcal{H}$. A transformer block $\text{Block}(\cdot): \mathbb{R}^{\hat{d}\times\tau} \to \mathbb{R}^{\hat{d}\times\tau}$ consists of a self-attention layer $\text{Att}(\cdot)$ and a feed-forward layer $\text{FF}(\cdot)$:

$$\text{Att}(\mathbf{h}_l) = \mathbf{h}_l + \sum_{i=1}^{h} W_O^i W_V^i \mathbf{h}_l \cdot \sigma[(W_K^i \mathbf{h}_l)^\top W_Q^i \mathbf{h}_l], \quad (2)$$

$$\begin{aligned}\mathbf{h}_{l+1} = \text{FF}(\mathbf{h}_l) &= \text{Att}(\mathbf{h}_l)\\ &+ W_2 \cdot \text{Relu}(W_1 \cdot \text{Att}(\mathbf{h}_l) + \mathbf{b_1}\mathbf{1}^\top) + \mathbf{b_2}\mathbf{1}^\top,\end{aligned} \quad (3)$$

with $W_O^i \in \mathbb{R}^{\hat{d}\times m_v}$, $W_V^i \in \mathbb{R}^{m_V\times\hat{d}}$, $W_K^i, W_Q^i \in \mathbb{R}^{m_h\times\hat{d}}$, $W_1 \in \mathbb{R}^{r\times\hat{d}}$, $W_2 \in \mathbb{R}^{\hat{d}\times r}$, $b_1 \in \mathbb{R}^r$ and $b_2 \in \mathbb{R}^{\hat{d}}$. Besides, $\mathbf{h}_l \in \mathbb{R}^{\hat{d}\times\tau}$ is the hidden state of $l$-th transformer block with $\mathbf{h}_0 \in \mathbb{R}^{\hat{d}\times\tau}$ is the input data $\mathcal{X} \in \mathbb{R}^{(\hat{d}_0\times\tau)\times n}$ (with position encoding) after a one-layer FCNN $F: \mathbb{R}^{\hat{d}_0\times\tau} \to \mathbb{R}^{\hat{d}\times\tau}$, and $\sigma(\cdot)$ is the column-wise softmax function. We denote $d := \hat{d}\times\tau$ and $d_0 := \hat{d}_0\times\tau$ for convenience. Formally, a classic transformer block with a budget of $m$ and $l$-th layer can be depicted as $\text{Block}_l^{(m)}(\cdot) = \text{FF}\circ\text{Att}(\cdot)$, where $m$ is the computational budget of a transformer block representing the width of transformers. LayerNorm is not taken into consideration because it does not affect the approximation capability of the transformer hypothesis space (Yun et al., 2019; Jiang & Li, 2024). Therefore, extending the hypothesis space in the theory to explicitly include LayerNorm leaves the analysis unchanged, which means including it does not change all of our theoretical results.

**Definition 2.1** (Budget of a transformer block).

$$m := (\hat{d}, h, m_h, m_V, r), \quad (4)$$

The computational budget of a transformer block $m$ includes the number of heads $h$, the hidden layer size $r$ of FF, $m_h$, $m_V$, and $n$ by the description of $\text{Block}_l^{(m)}(\cdot)$.

A transformer is a composition of transformer blocks, by which we define transformer hypothesis space $\mathcal{H}_{\text{Trans}}$:

**Definition 2.2** (Transformer hypothesis space). The transformer hypothesis space is $\mathcal{H}_{\text{Trans}}$ is

$$\mathcal{H}_{\text{Trans}} = \cup_l \mathcal{H}_{\text{Trans}}^{(l)} = \cup_l \cup_m \mathcal{H}_{\text{Trans}}^{(l,m)} \quad (5)$$

where $\mathcal{H}_{\text{Trans}}^{(l)}$ is the transformer hypothesis space with $l$ layers, and $\mathcal{H}_{\text{Trans}}^{(l,m)}$ is the transformer hypothesis space with $l$ layers of $\text{Block}_i^{(m)}(\cdot)$, $i \in \{1, 2, \ldots, l\}$. More specifically,

$$\mathcal{H}_{\text{Trans}}^{(l,m)} := \{\hat{H}: \hat{H} = \text{Block}_l^{(m)}\circ\cdots\circ\text{Block}_1^{(m)}\circ F\}. \quad (6)$$

The transformer hypothesis space encompasses all possible transformer configurations within a transformer neural network and serves as a foundational object of our study.

We design a classifier to distinguish between inlier and outlier data. By Definition 2.2 that $\forall \hat{H} \in \mathcal{H}_{\text{Trans}}$, $\hat{H}$ is a map from $\mathbb{R}^{d_0\times n}$ to $\mathbb{R}^{d\times n}$. To match the OOD detection task, we insert a classifier $c: \mathbb{R}^d \to \mathcal{Y}$ applied to each data as follows.

**Definition 2.3** (Classifier). $c: \mathbb{R}^d \to \mathcal{Y}$ is a classical classifier with forms:

(maximum value) $c(\mathbf{h}_l) = \arg\max_{k\in\mathcal{Y}} f^k(\mathbf{h}_l),$

(score-based) $c(\mathbf{h}_l) = \begin{cases} K+1, & E(f(\mathbf{h}_l)) < \lambda, \\ \arg\max_{k\in\mathcal{Y}} f^k(\mathbf{h}_l), & E(f(\mathbf{h}_l)) \geq \lambda, \end{cases}$

$(7)$

where $f^k$ is the $k$-th coordinate of $f \in \{\hat{f}: \mathbb{R}^d \to \mathbb{R}^{K+1}\}$, which is defined by

$$f^k(\mathbf{h}_l) = W_{4,k}(W_{3,k}\mathbf{h}_l + b_{3,k})^\top + b_{4,k}. \quad (8)$$

$W_{3,k} \in \mathbb{R}^{1\times\hat{d}}$, $W_{4,k}, b_{3,k} \in \mathbb{R}^{1\times\tau}$ and $b_{4,k} \in \mathbb{R}$. And $E(\cdot)$ is a scoring function like softmax-based function (Hendrycks & Gimpel, 2016) and energy-based function (Liu et al., 2020).

By combining Definitions 2.2 and 2.3, we can naturally derive the definition of the transformer hypothesis space for OOD detection as follows: a space that consists of all possible transformer models configured to classify and distinguish between inliers and outliers effectively.

**Definition 2.4** (Transformer hypothesis space for OOD detection).

$$\begin{aligned}\mathcal{H} := \{H \in \mathbb{R}^{d_0\times n} \to \mathcal{Y}^n : H = c\circ\hat{H}, \\ c \text{ is a classifier in Definition 2.3}, \hat{H} \in \mathcal{H}_{\text{Trans}}\}\end{aligned} \quad (9)$$

Similarly, we denote $\mathcal{H}^{(l)}$ as the transformer hypothesis space for OOD detection with exactly $l$ layers, and $\mathcal{H}^{(l,m)}$ with exactly $l$ layers and budget $m$ for each layer.

# 3. Theoretical Results

We focus on the learning theory of transformers within the four prior-unknown spaces (Fang et al., 2022): $\mathcal{D}_{XY}^{D_{XY}}$, $\mathcal{D}_{XY}^s$, $\mathcal{D}_{XY}^F$, and $\mathcal{D}_{XY}^{\mu,b}$. We do not exam the total space $\mathcal{D}_{XY}^{\text{all}}$ as the Impossible Theorem demonstrates that OOD detection is NOT learnable in this space, due to dataset overlap, even when the budget $m \to +\infty$. For each of the studied spaces, we investigate whether OOD detection is learnable under the transformer hypothesis space $\mathcal{H}$, taking into account the specific constraints or assumptions. When the learnability of OOD detection is established, we further analyze the approximation rates and error boundaries to gain deeper insights into the reliability of transformers.

## 3.1. OOD Detection in the Separate Space

When two target classes overlap, OOD detection struggles to accurately distinguish between them (Fang et al., 2022). Therefore, we focus exclusively on cases where the datasets of the two classes do not overlap, that is, corresponding to the separate space $\mathcal{D}_{XY}^s$. In this space, the absence of overlap allows for more effective learning and differentiation between inliers and outliers.

**Conditions for learning with transformers.** By Theorem 10 in Fang et al. (2022) and Theorems 5, 8 in Bartlett & Maass (2003), OOD detection is not learnable in $\mathcal{D}_{XY}^s$. So OOD detection is subject to the Impossible Theorem in $\mathcal{D}_{XY}^s$ for the transformer hypothesis space $\mathcal{H}$ unconditionally. We further explore the specific conditions required for $\mathcal{H}$ to achieve learnability. As a starting point, we derive a key lemma about the expressivity of transformers. The lemma establish the sufficient conditions under which transformers adhere to the universal approximation theorem, forming the theoretical basis for proving the learnability of OOD detection using transformers.

**Lemma 3.1.** *For any $\mathbf{h} \in \mathcal{C}(\mathbb{R}^d, \mathbb{R}^{K+1})$, and any compact set $C \in \mathbb{R}^d$, $\epsilon > 0$, there exists a two-layer transformer $\hat{\mathbf{H}} \in \mathcal{H}_{\text{Trans}}^{(m,2)}$ and a linear transformation $\mathbf{f}$ s.t. $\|\mathbf{f} \circ \hat{\mathbf{H}} - \mathbf{h}\|_2 < \epsilon$ in $C$, where $m = (K+1) \cdot \left( 2\tau(2\tau \hat{d}_0 + 1), 1, 1, \tau(2\tau \hat{d}_0 + 1), 2\tau(2\tau \hat{d}_0 + 1) \right)$.*

Lemma 3.1, derived as a corollary from the transformer approximation results of Jiang & Li (2024), tailors these findings to OOD detection tasks. Building on this, we establish sufficient and necessary conditions for OOD detection learnability on transformers with a fixed depth or width.

**Theorem 3.2** (Necessary and sufficient condition for OOD detection learnability on transformers)**.** *Given $l(\mathbf{y}_2, \mathbf{y}_1) \leq l(K+1, \mathbf{y}_1)$, for any in-distribution labels $\mathbf{y}_1, \mathbf{y}_2 \in \mathcal{Y}$, then OOD detection is learnable in the separate space $\mathcal{D}_{XY}^s$ for $\mathcal{H}$ if and only if $|\mathcal{X}| = n < +\infty$. Furthermore, if $|\mathcal{X}| < +\infty$, $\exists \delta > 0$ and $g \in \mathcal{H}^{(l,m)}$, where*

*Block$(\cdot)$ budget $m = (\hat{d}_0, 2, 1, 1, 4)$ and the number of Block$(\cdot)$ layer $l = \mathcal{O}\left( \tau(1/\delta)^{(\hat{d}_0 \tau)} \right)$, or $m = (K+1) \cdot \left( 2\tau(2\tau \hat{d}_0 + 1), 1, 1, \tau(2\tau \hat{d}_0 + 1), 2\tau(2\tau \hat{d}_0 + 1) \right)$ and $l = 2$ s.t. OOD detection is learnable with $g$.*

Theorem 3.2 provides a deeper understanding of transformers' capabilities and limitations for OOD detection. Detailed proof and remarks on inspection are in Appendix B.

**Extent of learnability by capacity of transformer network.** To quantify learnability as the budget $m$ grows, we obtain Jackson-type estimates for OOD detection learnability using transformer models, as established in Theorem 3.4 and Theorem 3.5. These estimates provide a theoretical framework to evaluate the quantitative relationship between model capacity and model learnability.

The extent of learnability of OOD detection can be defined as the probability that the algorithm can successfully learn the datasets and accurately recognize their class labels. The probability reflects the models's ability to generalize to unseen data, effectively distinguish inliers and outliers and correctly classify data points based on their underlying distribution. Formally, we define $\mathbf{P}$ as the probability of the learnable part in all data sets with $n$ data, when selecting the data subset in which the learnable data distribution accounts for the superior limit of the total data distribution.

**Definition 3.3** (Probability of the OOD detection learnability)**.** Given a domain space $\mathcal{D}_{XY}$ and the hypothesis space $\mathcal{H}^{(l,m)}$, $D_{XY}'^n \subset D_{XY}^n \in \mathcal{D}_{XY}$ is the distribution that for any dataset $\mathcal{X} \sim D_{X_I Y_I}'^n$, OOD detection is learnable, where $D_{XY}^n$ is any distribution in $\mathcal{D}_{XY}$ with data amount $n$. The probability of the OOD detection learnability is defined by

$$\mathbf{P} := \lim_{D_{XY}^n \in \mathcal{D}_{XY}} \overline{\lim_{D_{XY}'^n \subset D_{XY}^n}} \frac{\mu(D_{XY}'^n)}{\mu(D_{XY}^n)}, \tag{10}$$

where $\mu$ is the Lebesgue measure in $\mathbb{R}^d$ and $n \in \mathbb{N}^*$.

Theorem 3.4 and Theorem 3.5 of the Jackson-type approximation are formally expressed in terms of learnability probability as depicted in Definition 3.3. It reveals the precise relationship between model capacity and learnability for transformers in the OOD detection scenario, providing a rigorous framework to quantify how model size and structure influence the reliability of a transformer network in distinguishing inliers and outliers.

**Theorem 3.4.** *Given the condition $l(\mathbf{y}_2, \mathbf{y}_1) \leq l(K+1, \mathbf{y}_1)$, for any ID labels $\mathbf{y}_1, \mathbf{y}_2 \in \mathcal{Y}$, $|\mathcal{X}| = n < +\infty$ and $\tau > K+1$, and set $l = 2$ and $m = (2m_h + 1, 1, m_h, 2\tau \hat{d}_0 + 1, r)$. Then in $\mathcal{H}^{(l,m)}$ restricted to maximum value classifier $c$, $\mathbf{P} \geq (1 - \mathcal{O}(\frac{1}{m_h^{2\alpha-1}} + (\frac{m_h}{r})^\beta))^{(K+1)^{n+1}}$, where $\alpha$ and $\beta$ are constant from the regularity measures $C_1^{(\alpha)}$ and $C_2^{(\beta)}(\cdot)$.*

**Theorem 3.5.** *Given the condition as Theorem 3.4. In $\mathcal{H}^{(l,m)}$ restricted to score-based classifier $c$, $\mathbf{P} \geq (1 - \mathcal{O}(\frac{1}{m_h^{2\alpha-1}} + (\frac{m_h}{r})^\beta)^{(K+1)^{n+1}+1}$, if there exists $\lambda \in \mathbb{R}$ s.t. $\{\mathbf{v} \in \mathbb{R}^{K+1} : E(\mathbf{v}) \geq \lambda\}$ and $\{\mathbf{v} \in \mathbb{R}^{K+1} : E(\mathbf{v}) < \lambda\}$ both contain an open ball with the radius $R$, where $R > C(\tau^2 \mathcal{O}(\frac{1}{m_h^{2\alpha-1}} + (\frac{m_h}{r})^\beta) + \lambda_0), \forall \lambda_0 > 0, \exists C$ a constant.*

The proof employs the Jackson-type approximation for Transformers (Jiang & Li, 2024) to fulfill a sufficient condition for OOD detection learnability, namely Theorem 7 in Fang et al. (2022). Crucially, this Jackson-type approximation offers a global error bound, distinct from the uniform convergence typical of universal approximation property (UAP) theory (Jiang & Li, 2024), thereby necessitating Markov's inequality to derive probabilistic conclusions. This approach establishes a lower bound on the learning probability and its convergence rate for OOD detection using Transformers. It also unveils a scaling law: greater data complexity demands an increased number of parameters to maintain a sufficiently high learnable probability. The derived bound is not an infimum, as the Jackson-type approximation serves as a sufficient but not necessary condition (complete details are provided in Appendix C). Furthermore, based on Yun et al. (2019) and Remark B.3, our core conclusions (Theorems 3.2, 3.4, and 3.5) can be extended to more general Transformer architectures featuring larger budgets and depths. This signifies that Transformers beyond minimal configurations are also learnable under the same established theoretical conditions.

### 3.2. OOD Detection in Other Prior-Unknown Spaces

The remaining three prior-unknown spaces—the single-distribution space $\mathcal{D}_{XY}^{D_{XY}}$, the Finite-ID-distribution space $\mathcal{D}_{XY}^F$, and the density-based space $\mathcal{D}_{XY}^{\mu,b}$—do not require consideration if there exists an overlap between ID and OOD as OOD detection becomes unlearnable in such cases, as discussed in Fang et al. (2022). However, if the ID and OOD classes are non-overlapping, then since $\mathcal{D}_{XY}^{D_{XY}} \subset \mathcal{D}_{XY}^s$, the analysis has already been covered in the previous Section 3.1. Additionally, in the density-based space $\mathcal{D}_{XY}^{\mu,b}$, Theorem 9 and Theorem 11 in Fang et al. (2022) remain valid within the hypothesis space $\mathcal{H}$, as their proofs only need to verify the finite Natarajan dimension (Shalev-Shwartz & Ben-David, 2014) of the hypothesis space, which is a weaker condition than having the finite VC dimension.

## 4. Perspective of Leveraging Auxiliary Outliers

**Gap of theory and training** Theorems 3.2, 3.4, and 3.5 establish that models in $\mathcal{H}$ are learnable for OOD detection given sufficient parameters, offering a theoretical foundation for transformers in this task (Koner et al., 2021; Fort

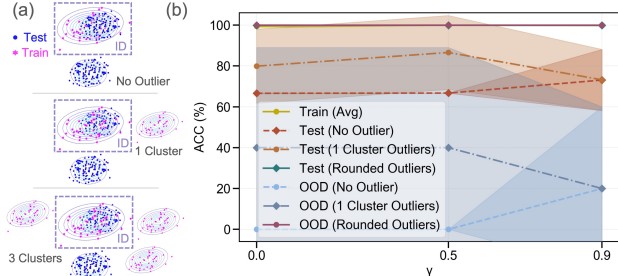

*Figure 1.* The more diverse outliers are, the better OOD detection performance transformers achieve. (a) Various distributions of ID and OOD. (b) Mean and standard deviation results under five random seeds.

et al., 2021). However, OOD detection remains challenging for transformers, for which we give two reasons in the theoretical perspective: (1) Theorems are under non-overlapping assumptions of ID-OOD distributions, but real-world OOD distributions may be ill-defined, making strict non-overlapping assumptions unachievable. This is an issue beyond algorithmic optimization. **(2)** The penalty for ID-OOD misclassification exceeds that for ID. But cross-entropy loss, commonly used for ID classification, does not penalize ID-OOD misclassification errors. This deviation from the theoretical assumptions gives an explanation for existing methods such as incorporating extra data (Fort et al., 2021; Tao et al., 2023) and using various distance metrics (Podolskiy et al., 2021).

To validate this inference, we first prove that with sufficient model depth and theoretical guarantees, an optimal OOD detection solution already exists in the parameter space by experiments, implying that detection errors are not due to the model's insufficient model capacity (see Appendix D). As shown in Fig. 1, experiments on Gaussian mixture data confirm that transformers trained solely with cross-entropy loss misclassify OOD as ID. Then, we refined the training paradigm by incorporating an ID-OOD binary classification loss and introducing a synthetic OOD data generation strategy, which gains optimal performance. Since the theorem does not assume access to real OOD datasets during training, we adopted a synthetic outlier generation approach, distinct from Outlier Exposure (OE) methods (Yang et al., 2024). This strategy enhances model robustness and reliability against unseen outliers, aligning with real-world scenarios where specific OOD samples may be unavailable in advance (Fort et al., 2021; Koner et al., 2021).

**Condition1: ID-OOD binary classification loss function.** First, considering that the classical cross-entropy loss $\mathcal{L}_1$ does not satisfy the condition $l(\mathbf{y}_2, \mathbf{y}_1) \leq l(K+1, \mathbf{y}_1)$, for any ID labels $\mathbf{y}_1, \mathbf{y}_2 \in \mathcal{Y}$, it provides no explicit instruction for models to recognize outliers. To address this, we

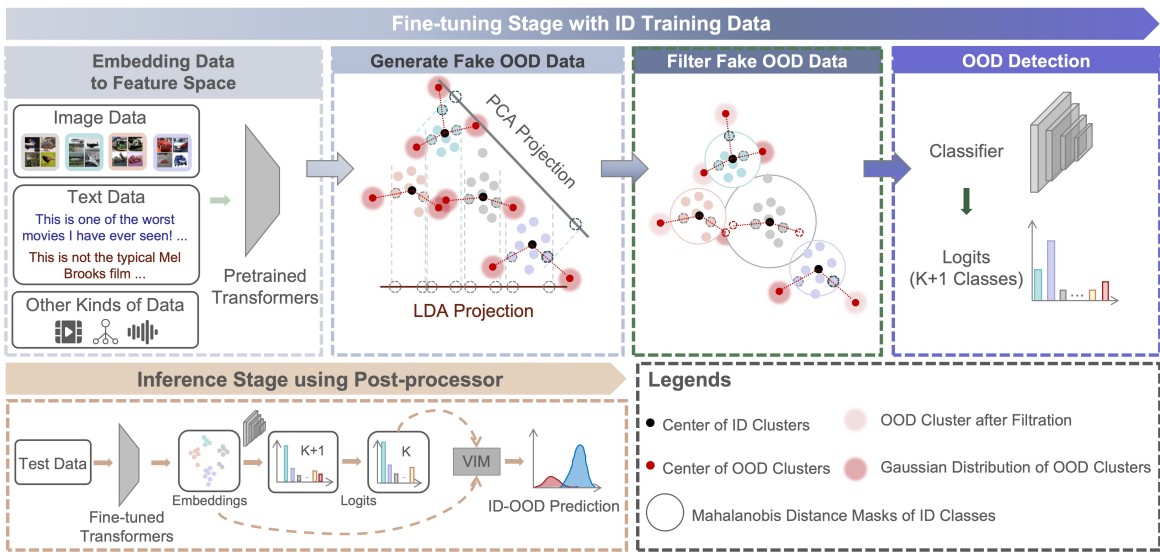

*Figure 2.* Overview of GROD: In fine-tuning, GROD generates fake OOD as part of the training data and guides fine-tuning by incorporating the ID-OOD loss $\mathcal{L}_2$. In inference, features and adjusted LOGITS are input into the post-processor.

---

**Algorithm 1** GROD

**Require:** $\mathcal{X}_{\text{train}}, \mathcal{Y}, \mathcal{X}_{\text{test}}, \mathcal{Y}_{\text{test}}$
**Ensure:** Trained model $M$, classification results $\hat{\mathcal{Y}}_{\text{test}}$
  {**Fine-tuning Stage**}
  **for** ep in training epochs **do**
    **for** each batch $\mathcal{X}$ in $\mathcal{X}_{\text{train}}$ **do**
      $\mathcal{F} \leftarrow \text{NET}(\mathcal{X})$ {Obtain features by Eq. (43)}
      Find boundary points $V_{\text{PCA}}$ and $V_{\text{LDA}}$ by Eq. (44)-Eq. (45)
      Generate fake OOD data $\hat{\mathcal{F}}^{\text{OOD}}$ by Eq. (46)-Eq. (52)
      Filter OOD data and get $\mathcal{F}^{\text{OOD}}$ by Eq. (53)-Eq. (55) and Random filtering mechanism
      Get soft labels $\mathcal{Y}^{\text{OOD}}$ for OOD data by Eq. (56)
      $\mathcal{F}_{\text{all}} \leftarrow \mathcal{F} \cup \mathcal{F}^{\text{OOD}}, \mathcal{Y}_{\text{all}} \leftarrow \mathcal{Y} \cup \mathcal{Y}^{\text{OOD}}$
      $\hat{\mathcal{Y}}_{\text{all}}, \text{LOGITS} \leftarrow \text{CLASSIFIER}(\mathcal{F}_{\text{all}})$
      Iterate the model parameters with $\mathcal{L}$ in Eq. (57)-(59).
    **end for**
    Save model $M$ with the best performance.
  **end for**
  {**Inference Stage**}
  $\mathcal{F}_{\text{test}}, \text{LOGITS}_{\text{test}} \leftarrow M(\mathcal{F}_{\text{test}})$
  $\text{LOGITS}_{\text{test}} \leftarrow \text{ADJUST}(\text{LOGITS}_{\text{test}})$ by Eq. (60)
  $\hat{\mathcal{Y}}_{\text{test}} \leftarrow \text{PostProcessor}(\mathcal{F}_{\text{test}}, \text{LOGITS}_{\text{test}})$
  Return $\hat{\mathcal{Y}}_{\text{test}}$

---

incorporate an additional loss term $\mathcal{L}_2$, into the overall loss:

$$\mathcal{L} = (1 - \gamma)\mathcal{L}_1 + \gamma\mathcal{L}_2, \tag{11}$$

$$\mathcal{L}_1(\mathbf{y}, \mathbf{x}) = - \underset{\mathbf{x} \in \mathcal{X}}{\mathbb{E}} \sum_{j=1}^{K+1} \mathbf{y}_j \log(\text{softmax}(\mathbf{f} \circ \mathbf{H}(\mathbf{x}))_j),$$

$$\mathcal{L}_2(\mathbf{y}, \mathbf{x}) = - \underset{\mathbf{x} \in \mathcal{X}}{\mathbb{E}} \sum_{j=1}^{2} \hat{\phi}(\mathbf{y})_j \log(\hat{\phi}(\text{softmax}(\mathbf{f} \circ \mathbf{H}(\mathbf{x})))_j),$$

$$\tag{12}$$

where $\mathbf{H} \in \mathcal{H}_{\text{Trans}}$, $\mathbf{y}$ is the label vector, $\hat{\phi} : \mathbb{R}^{K+1} \to \mathbb{R}^2$

is given by $\hat{\phi}(\mathbf{y}) = \left[\sum_{i=1}^{K} \mathbf{y}_i, \mathbf{y}_{K+1}\right]^\top$.

When the condition is satisfied, the classification loss sensitivity of ID classification decreases, potentially affecting the classification performance of ID. This suggests a trade-off in the choice of parameter $\gamma$ between ID data classification accuracy and OOD recognition, quantitatively observed in Fig. 1 selecting $\gamma = 0.0, 0.5, 0.9$. However, modifying the loss function without auxiliary outliers does not ensure stable training for OOD detection. This limitation arises because when the model correctly classifies ID data, $\mathbf{f} \circ \mathbf{H}(\mathbf{x})_{K+1}$ remains close to zero, rendering $\mathcal{L}_2$ ineffective and hindering the model's ability to distinguish ID from OOD. In the absence of OOD during training, the model is prone to misclassifying all test data as ID. So we explore the generation of virtual OOD data.

**Condition2: Generating rounded outliers.** Fig. 1 illustrates the accuracy on training and test sets when generating 0, 1, or 3 clusters of virtual OOD. As the generated OOD becomes more diverse, the model's performance in classification and OOD detection improves. When 3 clusters of rounded OOD are introduced, the model achieves optimal performance. This underscores the importance of generating high-quality virtual OOD to facilitate $\mathcal{L}_2$ and address the challenges posed by high-dimensional ID boundaries. Therefore, we provide a perspective for leveraging auxiliary outliers. For example, Fort et al. (2021) shows that incorporating outlier exposure significantly improves the OOD detection performance of transformers, while Du et al. (2022); Tao et al. (2023) synthesize outliers from ID samples.

## 5. GROD

Following Section 4, our theory identifies two necessary conditions for guaranteeing OOD learnability in GROD, namely **(C1)** the loss must penalize ID-OOD misclassification more heavily than within-ID errors (Theorem 3.2 and Jackson-type Theorems), and **(C2)** Auxiliary non-overlapping OOD data with high quality is required in training. We have designed GROD, which consists of several pivotal steps, as illustrated in Fig. 2 and Algorithm 1.

To address (C1), which requires heavier penalties for ID-OOD misclassification, we introduce an ID-OOD binary classification loss $\mathcal{L}_2$. This adjustment aligns more closely with the transformer's learnable conditions in the proposed Theorems 3.2, 3.4, and 3.5. We employ soft labeling because near-boundary synthetic samples vary in difficulty. This graded supervision better matches the asymmetric ID-OOD learning objective than hard binary labels and improves the robustness of the non-overlap condition across different distributions.

To address (C2), useful auxiliary OOD samples must lie close enough to the ID boundary to remain informative, while staying outside the ID region to ensure non-overlap. GROD is specifically designed around this trade-off to achieve high-quality OOD representations, with OOD generation and filtering methods. Instead of using raw data, GROD generates virtual OOD embeddings to minimize computational overhead while preserving rich feature representations. Principal Component Analysis (PCA) and Linear Discriminant Analysis (LDA) projections are employed to detect representative boundary ID, generating global and inter-class outliers, respectively, utilizing overall ID distribution and class-specific features. We then shift the ID boundaries outward to define the outlier centers. Mahalanobis filtering is then applied to explicitly remove generated points that lie too close to other ID clusters, directly satisfying the non-overlap requirement in our theory. We also employ autoregressive center updates to enable accurate ID estimation and stable distribution tracking during fine-tuning, which leads to more precise boundary estimation and better fulfills the non-overlap condition. We then fine-tune the transformer with datasets with virtual OOD with $\mathcal{L}$. During the testing phase, embeddings and prediction LOGITS are extracted from the GROD-enhanced transformer and reformulated for post-processing. A modified VIM (Wang et al., 2022) is applied to obtain the final prediction.

As defined in Definition 2.4, GROD gains a theoretical guarantee on transformers with multiple transformer layers and a classifier for OOD detection and classification tasks. So GROD has compatibility with transformers that extract features from the final layer, such as CLS tokens, before feeding them into the classifier—making GROD applicable to all transformer architectures. Formally, we also give the pseudocode of GROD displayed in Algorithm 1. Details of GROD method are provided in Appendix E. Moreover, the applicability and discussion of the proposed theory and algorithm GROD are in Appendix H.

## 6. Experiments

**Settings.** We use GROD to strengthen the reliability of self-supervised pre-trained transformers: DINO (Caron et al., 2021) for CV tasks, and encoder-only BERT (Devlin et al., 2018), decoder-only GPT-2 (Radford et al., 2019), and Llama-3.1-8B for NLP tasks. Metrics are FPR@95, AUROC, AUPR_IN, and AUPR_OUT. Baselines for comparison are MSP (Hendrycks & Gimpel, 2016), ODIN (Liang et al., 2017), VIM (Wang et al., 2022), GEN (Liu et al., 2023a), and ASH (Djurisic et al., 2022) which require only post-processing, and finetuning methods G-ODIN (Hsu et al., 2020), NPOS (Tao et al., 2023), and CIDER (Ming et al., 2022c). All baselines are based on OpenOOD benchmark (Zhang et al., 2023b; Yang et al., 2022a;b; 2021; Bitterwolf et al., 2023). Details are in Appendix F.

**Main results.** Regarding CV tasks, Table 1 and Table 2 summarize the comparative results of GROD on **CIFAR-10**, **CIFAR-100**, and **ImageNet-200** ID benchmarks. On **CIFAR-10**, GROD consistently outperforms all baselines, achieving a superior average FPR@95 of 3.01% and an AUROC of 99.33%. GROD demonstrates remarkable stability on **CIFAR-100**, where it reduces the average FPR@95 by a margin of at least 5.43% compared to all other competitors. On **ImageNet-200**, GROD achieves the lowest average FPR@95 of 30.90% and the highest average AUROC of 92.08%. For NLP tasks, we evaluate GROD across different transformer architectures and outlier types as shown in Table 3 and Table 6 (Appendix G), demonstrating its adaptability across different modalities. "-C" and "-L" denote with or without CLS tokens for LOGITS-based OOD detection. While text-based OOD detection remains a formidable challenge, GROD consistently outperforms existing approaches, achieving a reduction in FPR@95 of up to $10.68\%$ and maintaining superior performance across all OOD scenarios.

**Computational cost.** By appropriately selecting $|I|$ in Eq. (47), we ensure an effective fine-tuning that minimizes time costs while maximizing performance gains. In post-processing, we save the fine-tuned transformers without extra parameters, highlighting their computational advantages in real-world applications. Fig. 3 presents a quantitative comparison of the time costs. Combined with the superior task performance, it is evident that GROD achieves an optimal balance between effectiveness and efficiency. More results are available in Appendix G.

*Table 1.* OOD Detection Results (FPR@95 and AUROC) on ID datasets **CIFAR-10** and **CIFAR-100**. OOD dataset **CIFAR** represents **CIFAR-100** and **CIFAR-10** when ID datasets are **CIFAR-10** and **CIFAR-100** respectively.

| Methods | | *Evaluation under FPR@95 (%)* ↓ | | | | | | | *Evaluation under AUROC (%)*↑ | | | | | | |
| | | *Near-OOD* | | *Far-OOD* | | | | AVG | *Near-OOD* | | *Far-OOD* | | | | AVG |
| | | CIFAR | TIN | MNIST | SVHN | Texture | Places365 | | CIFAR | TIN | MNIST | SVHN | Texture | Places365 | |
| **ID: CIFAR-10** | | | | | | | | | | | | | | | |
| **Baseline** | MSP | 19.33 | 9.11 | 12.68 | 13.94 | 2.33 | 8.79 | 11.03 | 95.42 | 97.96 | 97.50 | 96.43 | 99.56 | 98.15 | 97.50 |
| **PostProcess** | ODIN | 77.87 | 56.06 | 54.09 | 61.62 | 62.50 | 79.38 | 65.25 | 52.49 | 64.33 | 88.00 | 77.10 | 77.28 | 77.02 | 72.70 |
| | VIM | 23.74 | 3.22 | 3.57 | 0.79 | **0.00** | 1.12 | 5.41 | 95.22 | 99.06 | 99.14 | 99.63 | 99.77 | 99.45 | 98.71 |
| | GEN | 16.39 | 3.56 | 5.21 | 6.93 | 0.27 | 3.26 | 5.94 | 96.70 | 99.19 | 98.90 | 98.58 | 99.92 | 99.27 | 98.76 |
| | ASH | 16.47 | 2.60 | 7.41 | 5.68 | 0.36 | 2.47 | 5.83 | 96.62 | 99.36 | 98.39 | 98.84 | 99.91 | 99.38 | 98.75 |
| **Finetuning** | G-ODIN | 76.30 | 25.61 | 36.03 | 42.64 | 0.67 | 2.47 | 30.62 | 67.57 | 93.78 | 92.03 | 86.52 | 99.87 | 99.52 | 89.88 |
| | NPOS | 45.26 | 40.36 | 45.76 | 18.31 | 23.27 | 39.43 | 35.40 | 85.85 | 87.77 | 86.43 | 95.89 | 94.55 | 86.73 | 89.54 |
| | CIDER | 21.63 | 12.43 | 9.99 | 0.26 | 2.99 | 10.90 | 9.70 | 95.54 | 97.09 | 97.94 | 99.93 | 99.36 | 97.40 | 97.88 |
| | **GROD** | **14.46** | **0.32** | **1.54** | 1.69 | **0.00** | **0.04** | **3.01** | **97.15** | **99.83** | **99.46** | **99.58** | **100.00** | **99.97** | **99.33** |
| **ID: CIFAR-100** | | | | | | | | | | | | | | | |
| **Baseline** | MSP | 58.32 | 29.83 | 75.72 | 30.29 | 8.71 | 26.01 | 38.15 | 82.32 | 93.58 | 65.40 | 92.45 | 98.33 | 95.26 | 87.89 |
| **PostProcess** | ODIN | 85.22 | 65.17 | 54.28 | 77.38 | 52.10 | 54.20 | 64.73 | 59.99 | 79.90 | 74.19 | 62.58 | 80.54 | 80.62 | 72.97 |
| | VIM | 55.37 | 18.53 | 55.31 | 15.91 | **0.03** | **0.53** | 24.28 | 81.66 | 96.15 | 79.11 | 94.80 | 99.97 | **99.82** | 90.34 |
| | GEN | 75.03 | 13.19 | 70.84 | 16.92 | 0.63 | 6.62 | 30.54 | 82.19 | 97.28 | 76.07 | 95.46 | **99.82** | 98.75 | 91.60 |
| | ASH | 74.22 | 12.22 | 72.66 | 17.48 | 0.69 | 6.13 | 30.57 | 82.38 | 96.23 | 73.63 | 95.10 | **99.82** | 98.86 | 91.00 |
| **Finetuning** | G-ODIN | 75.74 | 46.16 | 51.88 | 75.43 | 33.29 | 44.46 | 54.49 | 62.97 | 75.92 | 72.36 | 63.20 | 93.23 | 87.83 | 75.92 |
| | NPOS | 47.61 | 30.97 | 45.18 | 11.87 | 14.30 | 36.97 | 31.15 | **86.88** | 91.88 | 78.20 | 97.61 | 96.98 | 90.28 | 90.31 |
| | CIDER | 54.11 | 29.81 | 56.77 | **7.52** | 16.53 | 32.20 | 32.82 | 85.48 | 92.58 | 73.76 | **98.65** | 96.73 | 91.84 | 89.84 |
| | **GROD** | 49.19 | **5.33** | 39.89 | 10.56 | 0.62 | 7.50 | **18.85** | 84.90 | **98.81** | **85.15** | 96.82 | 99.71 | 97.34 | **93.79** |

*Table 2.* OOD Detection Results (FPR@95 and AUROC) on **ImageNet-200**.

| Methods | | *Evaluation under FPR@95 (%)* ↓ | | | | | | *Evaluation under AUROC (%)* ↑ | | | | | |
| | | *Near-OOD* | | *Far-OOD* | | | AVG | *Near-OOD* | | *Far-OOD* | | | AVG |
| | | SSB-hard | NINCO | iNaturalist | Texture | OpenImage-O | | SSB-hard | NINCO | iNaturalist | Texture | OpenImage-O | |
| **Baseline** | MSP | 69.34 | 47.08 | 21.72 | 33.84 | 33.01 | 41.00 | 79.26 | 86.46 | 82.86 | 94.69 | 92.53 | 87.16 |
| **PostProcess** | ODIN | 96.56 | 97.92 | 98.76 | 97.74 | 97.84 | 97.76 | 43.22 | 36.41 | 27.85 | 48.48 | 38.54 | 38.90 |
| | VIM | 71.02 | 40.93 | 14.97 | 15.47 | 22.86 | 33.05 | 81.03 | 89.06 | 94.92 | 96.81 | 94.06 | 91.18 |
| | GEN | 73.11 | 42.59 | **12.02** | 22.37 | 25.87 | 35.19 | 79.91 | 42.59 | **96.91** | 95.61 | 93.22 | 81.65 |
| | ASH | 73.27 | 42.99 | 12.13 | 22.63 | 25.82 | 35.37 | 79.82 | 87.71 | 96.90 | 95.63 | 93.32 | 90.68 |
| **Finetuning** | G-ODIN | 89.49 | 88.50 | 83.96 | 76.94 | 87.13 | 85.20 | 48.19 | 51.60 | 63.99 | 70.18 | 52.23 | 57.24 |
| | NPOS | 68.50 | 40.48 | 21.18 | **12.74** | 24.02 | 33.38 | 82.33 | 88.34 | 91.90 | 96.86 | 92.68 | 90.42 |
| | CIDER | 76.09 | **39.79** | 17.70 | 15.10 | 21.86 | 34.11 | 76.84 | 88.43 | 92.92 | 96.99 | 93.00 | 89.64 |
| | **GROD** | **65.82** | 39.87 | 14.90 | 13.79 | **20.13** | **30.90** | **83.54** | **90.10** | 95.20 | **97.12** | **94.42** | **92.08** |

*Table 3.* Quantitative comparison of NLP tasks on LORA (Hu et al., 2021) fine-tuned Llama-3.1-8B.

| OOD Type
ID Datasets
OOD Datasets | | Background Shift
**IMDB**
**Yelp** | | | | Semantic Shift
**CLINC150** with Intents
**CLINC150** with Unknown Intents | | | |
| *Evaluate Metrics (%)* | | FPR@95 ↓ | AUROC ↑ | AUPR_IN ↑ | AUPR_OUT ↑ | FPR@95 ↓ | AUROC ↑ | AUPR_IN ↑ | AUPR_OUT ↑ |
| **Baseline-L** | MSP-L | 96.90 | 46.21 | 36.04 | 58.82 | 31.76 | 91.13 | 97.50 | 70.00 |
| **Baseline-C** | MSP-C | 96.03 | 41.10 | 35.12 | 53.90 | 26.96 | 92.73 | 98.11 | 74.86 |
| **PostProcess** | VIM | 99.76 | 40.33 | 38.00 | 51.14 | 25.24 | 93.59 | 98.34 | 76.09 |
| | GEN-L | 96.90 | 46.21 | 36.04 | 58.82 | 28.84 | 93.18 | 97.93 | **79.35** |
| | GEN-C | 96.84 | 40.97 | 33.49 | 55.28 | 25.64 | 93.30 | 98.24 | 75.78 |
| | ASH | 93.96 | 57.66 | 44.90 | 65.86 | 27.16 | **93.69** | 98.30 | 77.65 |
| **Finetuning** | NPOS | 99.99 | 25.57 | 27.28 | 46.63 | 43.80 | 90.84 | 97.60 | 69.86 |
| | CIDER | 99.98 | 24.77 | 27.33 | 46.14 | 37.87 | 92.00 | 97.87 | 75.99 |
| | **GROD** | **83.28** | **67.00** | **57.70** | **72.42** | **21.60** | 93.64 | **98.55** | 76.19 |

# 7. Related Works

OOD detection has progressed significantly in methodologies and theoretical insights. Recent works improve performance through post-processing techniques, such as distance functions (Denouden et al., 2018), scoring functions (Ming et al., 2022a), and disturbance integration (Hsu et al., 2020), as well as training strategies, including compact loss functions (Tao et al., 2023) and anomaly reconstruction models (Graham et al., 2023; Jiang et al., 2023). Transformers are increasingly applied for OOD detection due to their robust feature representations (Koner et al., 2021; Fort et al., 2021). Auxiliary outliers are leveraged through Outlier Exposure

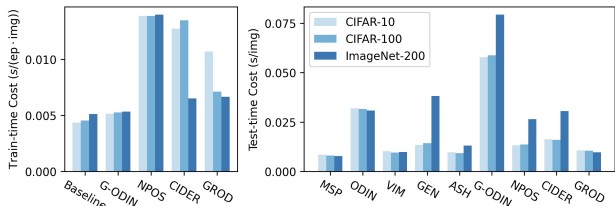

*Figure 3.* Comparison of the computational costs. Methods with only post-processing are employed after "Baseline" fine-tuning.

(OE) (Hendrycks et al., 2018; Zhu et al., 2023) or synthetic OOD data generation, such as VOS (Du et al., 2022) and OpenGAN (Kong & Ramanan, 2021), reducing reliance on predefined outliers (Wang et al., 2023b; Zheng et al., 2023). Theoretical contributions include works on maximum likelihood estimation (Morteza & Li, 2022), density estimation errors (Zhang et al., 2021), and PAC learning theory (Fang et al., 2022). However, transformer-specific OOD detection theory remains underdeveloped (Yang et al., 2021), limiting the reliability of algorithms. Details are in Appendix A.

## 8. Conclusion

We establish a PAC learning framework for OOD detection in transformers, providing necessary and sufficient conditions for learnability based on dataset distribution, training strategy, and model capacity. Additionally, we derive approximation rates and error bounds, offering theoretical insights to guide model selection and training design for reliable OOD detection. Building on these theoretical foundations, we propose a principled approach that synthesizes high-quality OOD representations using PCA, LDA, and Mahalanobis distance. This method fine-tunes transformer networks for more stable learning and is architecture-agnostic, making it broadly applicable across various tasks.

## Acknowledgements

This work is supported by Shanghai Jiao Tong University, Shanghai Innovation Institute, and the University of Oxford. And we would like to express our gratitude to our collaborators for their efforts.

## Impact Statement

This work provides a theoretical advancement for OOD detection for Transformer architectures, leveraging learning theory and approximation theory. These findings offer crucial, principled guidance for designing and analyzing more effective OOD detection mechanisms within widely used models. The primary impact will be the development of safer and more reliable AI systems capable of confidently handling novel inputs, which is critical as Transformers are increasingly deployed in sensitive, real-world applications, directly contributing to enhancing AI trustworthiness.

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

# A. Detailed Related Works.

**Application of OOD detection.**  The recent advancements in OOD detection models and algorithms have been significant (Sun et al., 2022; Liu et al., 2023b; Cai & Li, 2023). Typically, OOD detection methods leverage both post-processing techniques (Regmi, 2025) and training strategies (Tao et al., 2023), which can be implemented either separately or in combination (Zhang et al., 2023b). Key post-processing techniques include the use of distance functions (Denouden et al., 2018), the development of scoring functions (Ming et al., 2022a), and the integration of disturbance terms (Hsu et al., 2020), among others. Several methods introduce training strategies for OOD detection models. For instance, Tao et al. (2023) suggests loss functions to facilitate the learning of compact representations, while Graham et al. (2023); Jiang et al. (2023) innovatively employs reconstruction models to pinpoint abnormal data. In addition, the transformer architecture has gained popularity in OOD detection, prized for its robust feature representation capabilities (Koner et al., 2021; Fort et al., 2021; Liu & Qin, 2025).

**Leveraging auxiliary outliers.**  Leveraging auxiliary data for OOD detection has emerged as a prominent strategy. It is broadly categorized into Outlier Exposure (OE) and outlier-generating methods. OE involves utilizing external datasets as outliers during training to calibrate the model's ability to distinguish ID from OOD samples (Kirchheim & Ortmeier, 2022; Chen et al., 2021). Hendrycks et al. (Hendrycks et al., 2018) first proposed OE, demonstrating the effectiveness of using extra datasets, while Zhu et al. (Zhu et al., 2023) enhanced this method by introducing diversified outlier exposure through informative extrapolation. Zhang et al. (Zhang et al., 2023a) further extended this to fine-grained environments with Mixture Outlier Exposure, emphasizing the relevance of auxiliary outliers to specific tasks. ATOM (Chen et al., 2021) utilizes an adversarial outlier mining technique to pinpoint and select informative OOD samples that are crucial for effective model training. Similarly, Wang et al. (2023a) augments existing data distributions by meticulously optimizing auxiliary inputs in regions identified as potential OOD areas. Further contributing to data-centric strategies, DivOE (Zhu et al., 2023) promotes improved model extrapolation by systematically exposing the model to a wide array of diverse, synthetically generated outliers during its training phase. Another critical research thrust aims to improve the intrinsic mechanisms of OOD detection, particularly concerning uncertainty estimation. Illustrative of this, POEM (Ming et al., 2022b) employs posterior sampling techniques to achieve more robust and reliable uncertainty estimates, which are fundamental for accurately distinguishing OOD instances. Beyond training-time enhancements, adapting models during inference has also emerged as a key direction. AUTO (Yang et al., 2023), for example, introduces a framework that enables OOD detection mechanisms to adapt dynamically at test time, thereby allowing the model to better handle novel or shifting data characteristics encountered during deployment.

Generative-based methods, on the other hand, utilize generative models and feature modeling to create synthetic data that imitates OOD characteristics, thus enabling the generation of diverse and informative outlier samples without the need for predefined outlier datasets. VOS (Du et al., 2022) models the features as a Gaussian mixture distribution and samples out-of-distribution data in low-likelihood areas. NPOS (Tao et al., 2023) further uses KNN to generate out-of-distribution features. OpenGAN (Kong & Ramanan, 2021) pioneered this approach with GANs to generate open-set examples, and Wang et al. (Wang et al., 2023b) advanced it by employing implicit outlier transformations for more diverse OOD representations. Zheng et al. (Zheng et al., 2023) addressed scenarios with noisy or unreliable auxiliary data, refining generative processes for robust outlier synthesis. Du et al. (2024) is highlighted on generating high-resolution outliers in the pixel space using diffusion models. Furthermore, modeling soft labels is effective in presenting outliers and enhancing the decision-making connection between inliers and outliers (Lang et al., 2022; Xu et al., 2024), and it generalizes from OOD detection to other related fields such as toxicity classification (Cheng et al., 2024). These methods, by leveraging external or synthesized data, represent critical progress in enhancing OOD detection and improving model robustness in open-world scenarios.

**Theory of OOD detection.**  Theoretical research into OOD detection has recently intensified. Morteza & Li (2022) examines maximum likelihood on mixed Gaussian distributions and introduces a GEM log-likelihood score. Zhang et al. (2021) reveals that even minor errors in density estimation can result in OOD detection failures. Fang et al. (2022) presents the first application of Probably Approximately Correct (PAC) learning theory to OOD detection, deriving the Impossibility Theorem and exploring conditions under which OOD detection can be learned in previously unknown spaces. Moreover, Yang et al. (2021) has pioneered the concept of generalized OOD detection, noting its commonalities with anomaly detection (AD) and open set recognition (OSR) (Fang et al., 2021). To the best of our knowledge, no comprehensive theory of OOD detection for transformers has been established yet.

**Transformers and their universal approximation power**   Transformers bring inspiration and progress to OOD detection, with algorithms utilizing their self-attention mechanism achieving noteworthy results (Koner et al., 2021; Hendrycks et al., 2020; Podolskiy et al., 2021; Zhou et al., 2021). Understanding the expressivity of transformers is vital for their application in OOD detection. Current research predominantly explores two main areas: formal language theory and approximation theory (Strobl et al., 2024). The former examines transformers as recognizers of formal languages, clarifying their lower and upper bounds (Hahn, 2020; Chiang et al., 2023; Merrill & Sabharwal, 2024). Our focus, however, lies primarily in approximation theory. The universal approximation property (UAP) of transformers, characterized by fixed width and infinite depth, was initially demonstrated by Yun et al. (2019). Subsequent studies have expanded on this, exploring UAP under various conditions and transformer architectures (Yun et al., 2020; Kratsios et al., 2021; Luo et al., 2022; Alberti et al., 2023). As another important development, Jiang & Li (2024) established the UAP for architectures with a fixed depth and infinite width and provided Jackson-type approximation rates for transformers.

## B. Proof and Remarks of Theorem 3.2

In the data distribution spaces under our study, the equality of strong learnability and PAC learnability has been proved (Fang et al., 2022). So we only need to gain strong learnability to verify the proposed theorems. We first propose the lemma before proving Theorem 3.2.

**Lemma B.1.** *For any* $\mathbf{h} \in \mathcal{C}(\mathbb{R}^d, \mathbb{R}^{K+1})$, *and any compact set* $C \in \mathbb{R}^d$, $\epsilon > 0$, *there exists a two layer transformer* $\hat{\mathbf{H}} \in \mathcal{H}_{\mathrm{Trans}}^{(m,2)}$ *and a linear transformation* $\mathbf{f}$ *s.t.* $\|\mathbf{f} \circ \hat{\mathbf{H}} - \mathbf{h}\|_2 < \epsilon$ *in* $C$, *where* $m = (K+1) \cdot \left( 2\tau(2\tau\hat{d}_0 + 1), 1, 1, \tau(2\tau\hat{d}_0 + 1), 2\tau(2\tau\hat{d}_0 + 1) \right)$.

*Proof.* Let $\mathbf{h} = [h_1, \cdots, h_{K+1}]^\top$. Based on the UAP of transformers *i.e.* Theorem 4.1 in Jiang & Li (2024) (v1), for any $\epsilon > 0$, there exists $\hat{h}_i = \hat{f}_i \circ \bar{H}_i$, where $\hat{f}_i$ is a linear read out and $\bar{H}_i \in \mathcal{H}_{\mathrm{Trans}}^{(\hat{m},2)}$, $\hat{m} = 2\tau(2\tau\hat{d}_0+1), 1, 1, \tau(2\tau\hat{d}_0+1), 2\tau(2\tau\hat{d}_0+1)$ *s.t.*

$$\max_{\mathbf{x} \in C} \|\hat{h}_i(\mathbf{x}) - h_i(\mathbf{x})\|_1 < \epsilon/\sqrt{K+1}, i = 1, 2, \cdots, K+1. \tag{13}$$

We need to construct a transformer network $\hat{\mathbf{H}} \in \mathcal{H}_{\mathrm{Trans}}^{(m,2)}$ and a linear transformation $\mathbf{f}$ *s.t.*

$$(\mathbf{f} \circ \hat{\mathbf{H}})_i = \hat{f}_i \circ \bar{H}_i \tag{14}$$

for all $i \in \{1, \cdots, K+1\}$. The following shows the process of construction:

Denote the one-layer FCNN in $\bar{H}_i$ by $F_i : \mathcal{R}^{d_0 \times n} \to \mathcal{R}^{D \times n}$, where $D = 2n(2nd_0 + 1)$, the set the one-layer FCNN in $\hat{\mathbf{H}}$:

$$F : \mathcal{R}^{d_0 \times n} \to \mathcal{R}^{D(K+1) \times n},$$
$$F = [F_1, \cdots, F_{K+1}]^\top, \tag{15}$$

then $\mathbf{h}_0 = [h_0^1, \cdots, h_0^{K+1}]^\top$, where $\mathbf{h}_0$ is the input to transformer blocks in $\hat{\mathbf{H}}$, and $h_0^i$ is that in $\bar{H}_i$, $i = 1, \cdots, K+1$.

Denote the matrices in $\bar{H}_i$ by $\bar{W}_K^i$, $\bar{W}_Q^i$, $\bar{W}_V^i$ and $\bar{W}_O^i$ since each block only has one head. For the i-th head in each block of transformer network $\hat{\mathbf{H}}$, we derive the matrix $W_k^i \in \mathcal{R}^{(K+1)\hat{m}_h \times (K+1)D}$ from $\bar{W}_K^i$ with $\hat{m}_h = 1$:

$$W_K^i = \begin{bmatrix} \mathbf{0}_{(i-1)\hat{m}_h \times (i-1)D} & & \\ & \bar{W}_K^i & \\ & & \mathbf{0}_{(K+1-i)\hat{m}_h \times (K+1-i)D} \end{bmatrix}. \tag{16}$$

Furthermore, we obtain $W_Q^i$, $W_V^i$ and $W_O^i$ in the same way, then independent operations can be performed on different blocks in the process of computing the matrix $\mathrm{Att}(\mathbf{h}_0) \in \mathcal{R}^{(K+1)D \times n}$. So we can finally get the attention matrix in the following form:

$$\mathrm{Att}(\mathbf{h}_0) = [\mathrm{Att}_1(\mathbf{h}_0), \cdots \mathrm{Att}_{K+1}(\mathbf{h}_0)]^\top, \tag{17}$$

where $\mathrm{Att}_i(\mathbf{h}_0) \in \mathcal{R}^{D \times n}$, $i \in \mathcal{Y}_I + 1$ are attention matrices in $\bar{H}_i$.

Similarly, it is easy to select $W_1, W_2, \mathbf{b}_1, \mathbf{b}_2$ such that $\mathrm{FF}(\mathbf{h}_0) = [\mathrm{FF}_1(\mathbf{h}_0), \cdots \mathrm{FF}_{K+1}(\mathbf{h}_0)]^\top$, *i.e.* $\mathbf{h}_1 = [h_1^1, \cdots, h_1^{K+1}]^\top$, where the meaning of superscripts resembles to that of $h_0^i$. Repeat the process, we found that

$$\hat{\mathbf{H}}(\mathcal{X}) = [\bar{H}_1(\mathcal{X}), \cdots \bar{H}_{K+1}(\mathcal{X})]^\top. \tag{18}$$

Denote $\hat{f}_i(\bar{H}_i) = w_i \cdot \bar{H}_i + b_i$, then it is natural to construct the linear transformation $\mathbf{f}$ by:

$$\mathbf{f}(\hat{\mathbf{H}}) = [w_1, \cdots, w_{K+1}]^\top \cdot \hat{\mathbf{H}} + [b_1, \cdots, b_{K+1}]^\top, \tag{19}$$

which satisfies Eq. (14).

By Eq. (13), for any $\epsilon > 0$, there exists $\hat{\mathbf{H}} \in \mathcal{H}_{\text{Trans}}^{(m,2)}$ and the linear transformation $\mathbf{f}$ *s.t.*

$$\max_{\mathbf{x} \in C} \|\mathbf{f} \circ \hat{\mathbf{H}} - \mathbf{h}\|_2 \leq \sqrt{\sum_{i=1}^{K+1} (\max_{\mathbf{x} \in C} \|\hat{h}_i(\mathbf{x}) - h_i(\mathbf{x})\|_1)^2}$$
$$< \sqrt{\sum_{i=1}^{K+1} (\epsilon/\sqrt{K+1})^2} = \epsilon, \tag{20}$$

where $m = (K+1) \cdot \hat{m}$.

We have completed this Proof. □

Then we prove the proposed Theorem 3.2.

*Proof.* **First**, we prove the sufficiency. According to the Proof of Theorem 10 in Fang et al. (2022), to replace the FCNN-based or score-based hypothesis space with the transformer hypothesis space for OOD detection $\mathcal{H}$, the only thing we need to do is to investigate the UAP of transformer networks *s.t.* the UAP of FCNN network *i.e.* Lemma 12 in Fang et al. (2022) can be replaced by that of transformers. Moreover, it is easy to check Lemmas 13-16 in Fang et al. (2022) still hold for $\mathcal{H}$. So following the Proof of Theorem 10 in Fang et al. (2022), by Theorem 3 in Yun et al. (2019) and the proposed Lemma B.1, we can obtain the needed layers $l$ and specific budget $m$ which meet the conditions of the learnability for OOD detection tasks.

**Second**, we prove the necessity. Assume that $|\mathcal{X}| = +\infty$. By Theorems 5, 8 in Bartlett & Maass (2003), VCdim($\Phi \circ \mathcal{H}^{(l,m)}$) $< +\infty$ for any $m, l$, where $\Phi$ maps ID data to 1 and maps OOD data to 2. Additionally, $\sup_{h \in \mathcal{H}^{(l,m)}} |\{\mathbf{x} \in \mathcal{X} : h(\mathbf{x}) \in \mathcal{Y}\}| = +\infty$ given $|\mathcal{X}| = +\infty$ for any $m, l$. By the impossibility Theorem 5 for separate space in Fang et al. (2022), OOD detection is NOT learnable for any finite $m, l$. □

*Remark* B.2. Yun et al. (2019) and Jiang & Li (2024) (v1) provide two perspectives of the capacity of transformer networks. The former gives the learning conditions of OOD detection with limited width (or budget of each block) and any depth of networks, and the latter develops the learning conditions with limited depth.

*Remark* B.3. Define a partial order for the budget $m$: for $m = (d, h, m_h, m_V, r)$ and $m' = (d', h', m'_h, m'_V, r')$, $m' < m$ if every element in $m'$ is less than the corresponding element in $m$. $m' \leq m$ if if every element in $m'$ is not greater than the corresponding element in $m$. So it easily comes to a corollary: $\forall m'$ satisfies $m \leq m'$ and $l \leq l'$, if transformer hypothesis space $\mathcal{H}^{(l,m)}$ is OOD detection learnable, then $\mathcal{H}^{(m',l')}$ is OOD detection learnable.

*Remark* B.4. It is notable that when $m = +\infty$ or $l = +\infty$, VCdim($\Phi \circ \mathcal{H}^{(l,m)}$) may equal to $+\infty$. This suggests the possibility of achieving learnability in OOD detection without the constraint of $|\mathcal{X}| < +\infty$. Although an infinitely capacitated transformer network does not exist in reality, exploring whether the error asymptotically approaches zero as capacity increases remains a valuable theoretical inquiry.

## C. Proof and Remarks of Theorem 3.4 and Theorem 3.5

Firstly, we give the formal description of Theorem 3.4 and Theorem 3.5, integrating the two into Theorem C.1:

**Theorem C.1.** *Given the condition $l(\mathbf{y}_2, \mathbf{y}_1) \leq l(K+1, \mathbf{y}_1)$, for any in-distribution labels $\mathbf{y}_1, \mathbf{y}_2 \in \mathcal{Y}$, $|\mathcal{X}| = n < +\infty$ and $\tau > K + 1$, and set $l = 2$ and $m = (2m_h + 1, 1, m_h, 2\tau\hat{d}_0 + 1, r)$. Then in $\mathcal{H}_{\text{tood}}^{(m,l)}$ restricted to maximum value classifier $c$, $\mathbf{P} \geq (1 - \frac{\eta}{|\mathcal{I}|\lambda_0} \tau^2 C_0(r_i)(\frac{C_1^{(\alpha)}(r_i)}{m_h^{2\alpha-1}} + \frac{C_2^{(\beta)}(r_i)}{r^\beta}(km_h)^\beta))^{(K+1)^{n+1}}$, and in $\mathcal{H}_{\text{tood}}^{(m,l)}$ restricted to score-based classifier $c$, $\mathbf{P} \geq (1 - \frac{\eta}{|\mathcal{I}|\lambda_0} \tau^2 C_0(r_i)(\frac{C_1^{(\alpha)}(r_i)}{m_h^{2\alpha-1}} + \frac{C_2^{(\beta)}(r_i)}{r^\beta}(km_h)^\beta))^{(K+1)^{n+1}+1}$, for any fixed $\lambda_0 > 0$ and $r_i$ defined in*

*Lemma C.5, if $\{\mathbf{v} \in \mathbb{R}^{K+1} : E(\mathbf{v}) \geq \lambda\}$ and $\{\mathbf{v} \in \mathbb{R}^{K+1} : E(\mathbf{v}) < \lambda\}$ both contain an open ball with the radius $R$, where $R > ||W_4||_2 |\mathcal{I}| (\tau^2 C_0(\phi)(\frac{C_1^{(\alpha)}(\phi)}{m_h^{2\alpha-1}} + \frac{C_2^{(\beta)}(\phi)}{r^\beta}(km_h)^\beta) + \lambda_0)$, $\phi$ defined in Lemma C.6 and $W_4$ is determined by $\phi$.*

To derive Theorem 3.4 and Theorem 3.5, it is equivalent to prove Theorem C.1. We need to figure out some lemmas before deriving the theorem.

**Lemma C.2.** *For any $\mathbf{h} \in \widetilde{\mathcal{C}}^{(\alpha,\beta)}$, and any compact set $C \in \mathbb{R}^d$, there exists a two layer transformer $\hat{\mathbf{H}} \in \mathcal{H}_{\text{Trans}}^{(m,2)}$ and a linear read out $\mathbf{c} : \mathbb{R}^{\hat{d} \times \tau} \to \mathbb{R}^{1 \times \tau}$ s.t. the inequality (24) is established, where $m = (2m_h + 1, 1, m_h, 2\tau \hat{d}_0 + 1, r)$.*

*Proof.* According to Theorem 4.2 in Jiang & Li (2024), for any $\mathbf{h} \in \widetilde{\mathcal{C}}^{(\alpha,\beta)}$, there exists $\mathbf{H} \in \mathcal{H}_{\text{Trans}}^{(m,2)}$ and a linear read out $\mathbf{c}$ s.t.

$$\int_{\mathcal{I}} \sum_{t=1}^\tau |\mathbf{c} \circ \mathbf{H}_t(\mathbf{x}) - \mathbf{h}_t(\mathbf{x})| d\mathbf{x} \leq \tau^2 C_0(\mathbf{h}) \left( \frac{C_1^{(\alpha)}(\mathbf{h})}{m_h^{2\alpha-1}} + \frac{C_2^{(\beta)}(\mathbf{h})}{m_{\text{FF}}^\beta}(m_h)^\beta \right), \tag{21}$$

where $\mathcal{I}$ is the range of the input data. Thus

$$P\left( \sum_{t=1}^\tau |\mathbf{c} \circ \mathbf{H}_t(\mathbf{x})_i - \mathbf{h}_t(\mathbf{x})_i| / |\mathcal{I}| > \text{RHS in Eq. (21)} + \lambda_0 \right) \leq \frac{\text{RHS in Eq. (21)}}{\lambda_0 |\mathcal{I}|} \tag{22}$$

for any $\lambda_0 > 0$. Additionally,

$$\begin{aligned}
\left\| \mathbf{c} \circ \mathbf{H}(\mathbf{x}) - \mathbf{h}(\mathbf{x}) \right\|_2 &= \sqrt{\sum_{t=1}^\tau |\mathbf{c} \circ \mathbf{H}_t(\mathbf{x})_i - \mathbf{h}_t(\mathbf{x})_i|^2} \\
&\leq \sum_{t=1}^\tau |\mathbf{c} \circ \mathbf{H}_t(\mathbf{x})_i - \mathbf{h}_t(\mathbf{x})_i|.
\end{aligned} \tag{23}$$

So we get

$$P(\|\mathbf{c} \circ \mathbf{H}(\mathbf{x}) - \mathbf{h}(\mathbf{x})\|_2 > |\mathcal{I}|(\text{RHS in Eq. (21)} + \lambda_0)) \leq \frac{\text{RHS in Eq. (21)}}{\lambda_0 |\mathcal{I}|} \tag{24}$$

where $m_{\text{FF}}$ is usually determined by its number of neurons and layers. As the number of layers in FF is fixed, the budget $m_{\text{FF}}$ and $r$ are proportional with constant $k$:

$$r = k \cdot m_{\text{FF}}. \tag{25}$$

So the right side of the equation (21) can be written as

$$\text{RHS} = \tau^2 C_0(\mathbf{h}) \left( \frac{C_1^{(\alpha)}(\mathbf{h})}{m_h^{2\alpha-1}} + \frac{C_2^{(\beta)}(\mathbf{h})}{r^\beta}(km_h)^\beta \right). \tag{26}$$

We have completed this Proof of the Lemma C.2. $\qquad\square$

Given any finite $\delta$ hypothesis functions $h_1, \cdots, h_\delta \in \{\mathcal{X} \to \mathcal{Y}\}$, for each $h_i$, we introduce a correspongding $\mathbf{g}_i$ (defined over $\mathcal{X}$) satisfying that for any $\mathbf{x} \in \mathcal{X}$, $\mathbf{g}_i(\mathbf{x}) = \mathbf{y}_k$ and $W_4 \mathbf{g}_i^\top + b_4 = \mathbf{z}_k$ if and only if $h_i(\mathbf{x}) = k$, where $\mathbf{z}_k \in \mathbb{R}^{K+1}$ is the one-hot vector corresponding to the label $k$ with value $N$. Clearly, $\mathbf{g}_i$ is a continuous mapping in $\mathcal{X}$, because $\mathcal{X}$ is a discrete set. Tietze Extension Theorem (Urysohn, 1925) implies that $\mathbf{g}_i$ can be extended to a continuous function in $\mathbb{R}^d$. If $\tau \geq K + 1$, we can find such $\mathbf{g}_i, W_4, b_4$.

**Lemma C.3.** *For any introduced $\mathbf{g}_i$ mentioned above, there exists $\hat{\mathbf{g}}_i$ satisfies $\hat{\mathbf{g}}_i \in \widetilde{\mathcal{C}}^{(\alpha,\beta)}$ and $\|\hat{\mathbf{g}}_i - \mathbf{g}_i\|_2 < \epsilon$.*

*Proof.* Based on Theorem 7.4 in DeVore et al. (2021), set $G \equiv 0$ and $\rho \equiv 0$, then $\hat{\mathbf{g}}_i \in \widetilde{\mathcal{C}}^{(\alpha,\beta)}$, and there exists a constant $C$, s.t. $\|\hat{\mathbf{g}}_i - \mathbf{g}_i\|_2 < \frac{C}{(r+1)^\beta}$.

Choose $r$ which is great enough, the proof is completed. $\qquad\square$

*Remark* C.4. Note that we can also prove the same result if $\mathbf{g}_i$ is any continuous function from $\mathbb{R}^{\hat{d}}$ to $\mathbb{R}$ with compact support.

**Lemma C.5.** *Assume the labels are independent for classification. Let $|\mathcal{X}| = n < +\infty$, $\tau > K + 1$ and $\sigma$ be the* Relu *function. Given any finite $\delta$ hypothesis functions $h_1, \cdots, h_\delta \in \{\mathcal{X} \to \{1, \cdots, K + 1\}\}$, then for any $m_h, r > 0$, $m = (2m_h + 1, 1, m_h, 2\tau\hat{d}_0 + 1, r)$, $P(h_1, \cdots, h_\delta \in \mathcal{H}^{(m,2)}) \geq (1 - \frac{mRHS \text{ in Eq. (21)}}{|\mathcal{I}|\lambda_0})^{(K+1)\delta}$ for any $\eta > 1$.*

*Proof.* Since $\mathcal{X}$ is a compact set, then Lemma C.3 implies that there exists $\hat{\mathbf{g}}_i \in \widetilde{\mathcal{C}}^{(\alpha,\beta)}$ s.t.

$$\|\mathbf{g}_i - \hat{\mathbf{g}}_i\|_2 < \epsilon/\|W_4\|_2. \tag{27}$$

Denote $r_i = W_4\mathbf{g}_i^\top + b_4$ and $\hat{r}_i = W_4\hat{\mathbf{g}}_i^\top + b_4$, So we get

$$\|r_i - \hat{r}_i\|_2 = \|W_4(\mathbf{g}_i - \hat{\mathbf{g}}_i)^\top\|_2 \leq \epsilon. \tag{28}$$

Then by Lemma C.2, there exists $\hat{\mathbf{H}} \in \mathcal{H}_{\text{Trans}}^{(m,2)}$ and a linear read out $\mathbf{c}$ s.t.

$$P(\|\mathbf{c} \circ \mathbf{H}(\mathbf{x}) - \mathbf{h}(\mathbf{x})\|_2 \leq |\mathcal{I}|(\text{RHS in Eq. (21)} + \lambda_0) \geq 1 - \frac{\text{RHS in Eq. (21)}}{\lambda_0|\mathcal{I}|}. \tag{29}$$

Thus we get if $h_i(\mathbf{x}) = k$, which is equal to $\mathbf{g}_i(\mathbf{x}) = \mathbf{y}_k$ or $r_i(\mathbf{x}) = \mathbf{z}_k$:

Firstly, denote $\mathbf{f} = W_4\mathbf{c} \circ \mathbf{H}^\top + b_4$, and let $\mathbf{h} = \hat{\mathbf{g}}_i$, then

$$P(\|\mathbf{f}(\mathbf{x}) - \hat{r}_i(\mathbf{x})\|_2 \leq \|W_4\|_2|\mathcal{I}|(\text{RHS in Eq. (21)} + \lambda_0)) \geq 1 - \frac{\text{RHS in Eq. (21)}}{\lambda_0|\mathcal{I}|}. \tag{30}$$

So we obtain that

$$\begin{aligned}
P(|\mathbf{f}_k - N| &\leq \|W_4\|_2|\mathcal{I}|(\text{RHS in Eq. (21)} + \lambda_0)) \\
&\geq P(|\mathbf{f}_k - \hat{r}_{i,k}| + |\hat{r}_{i,k} - r_{i,k}| \leq \|W_4\|_2|\mathcal{I}|(\text{RHS in Eq. (21)} + \lambda_0)) \\
&\geq P(\|\mathbf{f} - \hat{r}_i\|_2 + \|\hat{r}_i - r_i\|_2 \leq \|W_4\|_2|\mathcal{I}|(\text{RHS in Eq. (21)} + \lambda_0)) \\
&\geq P(\|\mathbf{f} - \hat{r}_i\|_2 + \epsilon \leq \|W_4\|_2|\mathcal{I}|(\text{RHS in Eq. (21)} + \lambda_0)) \\
&= P\left(\|\mathbf{f} - \hat{r}_i\|_2 \leq \|W_4\|_2|\mathcal{I}|\left(\text{RHS in Eq. (21)} + (\lambda_0 - \frac{\epsilon}{|\mathcal{I}|}))\right)\right) \\
&\geq 1 - \frac{\text{RHS in Eq. (21)}}{|\mathcal{I}|(\lambda_0 - \frac{\epsilon}{|\mathcal{I}|})} \\
&= 1 - \frac{\text{RHS in Eq. (21)}}{|\mathcal{I}|\lambda_0 - \epsilon}.
\end{aligned} \tag{31}$$

Similarly, for any $j \neq k$, we can also obtain that

$$P(|\mathbf{f}_k| \leq \|W_4\|_2|\mathcal{I}|(\text{RHS in Eq. (21)} + \lambda_0) \geq 1 - \frac{\text{RHS in Eq. (21)}}{|\mathcal{I}|\lambda_0 - \epsilon}. \tag{32}$$

Therefore, $P(\arg\max_{k \in \mathcal{Y}} \mathbf{f}_k(\mathbf{x}) = h_i(\mathbf{x})) \geq (1 - \frac{\eta\text{RHS in Eq. (21)}}{|\mathcal{I}|\lambda_0})^{K+1}$ for any $\mathbf{x}$, if

$$N > 2\|W_4\|_2|\mathcal{I}|(\text{RHS in Eq. (21)} + \lambda_0) \tag{33}$$

for any $\eta > 1$, *i.e.*

$$P(h_1, \cdots, h_\delta \in \mathcal{H}^{(m,2)}) \geq \left(1 - \frac{\eta\text{RHS in Eq. (21)}}{|\mathcal{I}|\lambda_0}\right)^{(K+1)\delta}, \tag{34}$$

if

$$N > 2\|W_4\|_2|\mathcal{I}|(\text{RHS in Eq. (21)} + \lambda_0) \tag{35}$$

for any $\eta > 1$. Since $N$ is arbitrary, we can find such $N$. $\qquad\square$

**Lemma C.6.** *Let the activation function $\sigma$ be the Relu function. Suppose that $|\mathcal{X}| < +\infty$, and $\tau > K + 1$. If $\{\mathbf{v} \in \mathbb{R}^{K+1} : E(\mathbf{v}) \geq \lambda\}$ and $\{\mathbf{v} \in \mathbb{R}^{K+1} : E(\mathbf{v}) < \lambda\}$ both contain an open ball with the radius $R > \|W_4\|_2|\mathcal{I}|(RHS \text{ in Eq. (21)}(\phi) + \lambda_0)$, the probability of introduced binary classifier hypothesis space $\mathcal{H}_E^{(m,2),\lambda}$ consisting of all binary classifiers $P > (1 - \frac{\eta RHS \text{ in Eq. (21)}}{|\mathcal{I}|\lambda_0})^{(K+1)\delta+1}$, where $m = (2m_h + 1, 1, m_h, 2\tau\hat{d}_0 + 1, r)$ and $\phi(\mathbf{x})$ is determined by centers of balls, specifically defined in the proof and $W_4$ is determined by $\phi(\mathbf{x})$.*

*Proof.* Since $\{\mathbf{v} \in \mathbb{R}^{K+1} : E(\mathbf{v}) \geq \lambda\}$ and $\{\mathbf{v} \in \mathbb{R}^{K+1} : E(\mathbf{v}) < \lambda\}$ both contain an open ball with the radius $R \geq \|W_4\|_2|\mathcal{I}|(RHS \text{ in Eq. (21)} + \lambda_0)$, we can find $\mathbf{v}_1 \in \{\mathbf{v} \in \mathbb{R}^{K+1} : E(\mathbf{v}) \geq \lambda\}$, $\mathbf{v}_2 \in \{\mathbf{v} \in \mathbb{R}^{K+1} : E(\mathbf{v}) < \lambda\}$ *s.t.* $B_R(\mathbf{v}_1) \subset \{\mathbf{v} \in \mathbb{R}^{K+1} : E(\mathbf{v}) \geq \lambda\}$ and $B_R(\mathbf{v}_2) \subset \{\mathbf{v} \in \mathbb{R}^{K+1} : E(\mathbf{v}) < \lambda\}$, where $B_R(\mathbf{v}_1) := \{\mathbf{v} : \|\mathbf{v} - \mathbf{v}_1\|_2 < R\}$ and $B_R(\mathbf{v}_2) := \{\mathbf{v} : \|\mathbf{v} - \mathbf{v}_2\|_2 < R\}$.

For any binary classifier $h$ over $\mathcal{X}$, we can induce a vector-valued function as follows. For any $\mathbf{x} \in \mathcal{X}$,

$$\phi(\mathbf{x}) = \begin{cases} \mathbf{v}_1, & \text{if } h(\mathbf{x}) = 1, \\ \mathbf{v}_2, & \text{if } h(\mathbf{x}) = 2. \end{cases} \tag{36}$$

Since $\mathcal{X}$ is a finite set, the Tietze Extension Theorem implies that $\phi$ can be extended to a continuous function in $\mathbb{R}^d$. Since $\mathcal{X}$ is a compact set, then Lemma C.2 and Lemma C.3 implies that there exists a two layer transformer $\mathbf{H} \in \mathcal{H}_{\text{Trans}}^{(m,2)}$ and $f$ defined in Definition 2.3 s.t for any $\eta > 1$,

$$P(\|\mathbf{f} \circ \mathbf{H}(\mathbf{x}) - \phi(\mathbf{x})\|_2 \leq \|W_4\|_2|\mathcal{I}|(RHS \text{ in Eq. (21)} + \lambda_0) \geq 1 - \frac{RHS \text{ in Eq. (21)}}{|\mathcal{I}|\lambda_0 - \epsilon} \tag{37}$$

Therefore, for any $\mathbf{x} \in \mathcal{X}$, it is easy to check that $E(\mathbf{f} \circ \mathbf{H}(\mathbf{x})) \geq \lambda$ if and only if $h(\mathbf{x}) = 1$, and $E(\mathbf{f} \circ \mathbf{H}(\mathbf{x})) < \lambda$ if and only if $h(\mathbf{x}) = 2$ if the condition in $P(\cdot)$ is established.

Since $|X| < +\infty$, only finite binary classifiers are defined over $\mathcal{X}$. By Lemma C.5, we get

$$P(\mathcal{H}_{\text{all}}^b = \mathcal{H}_E^{(m,2),\lambda}) \geq \left(1 - \frac{\eta RHS \text{ in Eq. (21)}}{|\mathcal{I}|\lambda_0}\right)^{(K+1)\delta+1} \tag{38}$$

The proof is completed. $\qquad\square$

Now we prove one of the main conclusions *i.e.* Theorem 3.4 and Theorem 3.5, which provides a sufficient Jackson-type condition for learning of OOD detection in $\mathcal{H}$.

*Proof.* **First, we consider the case that $c$ is a maximum value classifier.** Since $|\mathcal{X}| < +\infty$, it is clear that $|\mathcal{H}_{\text{all}}| < +\infty$, where $\mathcal{H}_{\text{all}}$ consists of all hypothesis functions from $\mathcal{X}$ to $\mathcal{Y}$. For $|\mathcal{X}| < +\infty$ and $\tau > K + 1$, according to Lemma C.5, $P(\mathcal{H}_{\text{all}} \subset \mathcal{H}^{(m,2)}) \geq (1 - \frac{\eta RHS \text{ in Eq. (21)}}{|\mathcal{I}|\lambda_0})^{(K+1)\delta}$ for any $\eta > 1$, where $m = (2m_h + 1, 1, m_h, 2nd + 1, r)$ and $\delta = (K+1)^n$.

Consistent with the proof of Lemma 13 in Fang et al. (2022), we can prove the correspondence Lemma 13 in the transformer hypothesis space for OOD detection if $\mathcal{H}_{\text{all}} \subset \mathcal{H}^{(m,2)}$, which implies that there exist $\mathcal{H}^{\text{in}}$ and $\mathcal{H}^b$ *s.t.* $\mathcal{H}^{(m,2)} \subset \mathcal{H}^{\text{in}} \circ \mathcal{H}^b$, where $\mathcal{H}^{\text{in}}$ is for ID classification and $\mathcal{H}^b$ for ID-OOD binary classification. So it follows that $\mathcal{H}_{\text{all}} = \mathcal{H}^{(m,2)} = \mathcal{H}^{\text{in}} \circ \mathcal{H}^b$. Therefore, $\mathcal{H}_b$ contains all binary classifiers from $\mathcal{X}$ to $\{1, 2\}$. According to Theorem 7 in (Fang et al., 2022), OOD detection is learnable in $\mathcal{D}_{XY}^s$ for $\mathcal{H}^{(m,2)}$.

**Second, we consider the case that $c$ is a score-based classifier.** It is easy to figure out the probability of which OOD detection is learnable based on Lemma C.6 and Theorem 7 in Fang et al. (2022).

The proof of Theorem 3.4 and Theorem 3.5 is completed. $\qquad\square$

*Remark* C.7. Approximation of $\alpha$: First of all, it is definitely that $\alpha > \frac{1}{2}$ to maintain the conditions in Theorem 4.2 of Jiang & Li (2024). Then, analyze the process of our proof, because of the powerful expressivity of Relu, we only need $G \equiv 0$ to bridge from $\mathcal{C}$ to $\widetilde{\mathcal{C}}^{(\alpha,\beta)}$. So with regard to $\mathcal{H}$, any $\alpha > \frac{1}{2}$ satisfies all conditions. But $C_1^\alpha$ can increase dramatically when $\alpha$ get greater.

*Remark* C.8. Approximation of $\beta$: We denote $\beta \in (0, \beta_{\max}]$. According to Theorem 7.4 in DeVore et al. (2021), $\beta_{\max} \in [1, 2]$.

*Remark* C.9. By the approximation of $\alpha$ and $\beta$, we discuss the trade-off of expressivity and the capacity of transformer models. Firstly, the learnability probability $P \to 1$ if and only if $m_h \to +\infty$ and $\frac{r}{m_h} \to +\infty$. For a fixed $r$, there exists a $m_h$ which achieves the best trade-off. For a fixed $m_h$, the greater $r$ is, the more powerful the expressivity of transformer models is.

*Remark* C.10. Different scoring functions $E$ have different ranges. For example, $\max_{k \in \{1, \cdots K\}} \frac{e^{v^k}}{\sum_{c=1}^{K+1} e^{v^c}}$ and $T \log \sum_{c=1}^{K} e^{(\frac{v^c}{T})}$ have ranges contain $(\frac{1}{K+1}, 1)$ and $(0, +\infty)$, respectively. Theorem 3.4 and Theorem 3.5 give the insight that the domain and range of scoring functions should be considered when dealing with OOD detection tasks using transformers.

*Remark* C.11. It can be seen from Theorem 3.4 and Theorem 3.5 that the complexity of the data increases, and the scale of the model must also increase accordingly to ensure the same reliability from the perspective of OOD detection. Increasing the category $K$ of data may exponentially reduce the learnable probability of OOD detection, while increasing the amount of data $n$ reduces the learnable probability much more dramatically. Using Taylor expansion for estimation,

$$
\begin{aligned}
&\left(1 - \frac{\eta}{|\mathcal{I}|\lambda_0} \tau^2 C_0(r_i)\big(\frac{C_1^{(\alpha)}(r_i)}{m_h^{2\alpha-1}} + \frac{C_2^{(\beta)}(r_i)}{r^\beta}(km_h)^\beta\big)\right)^{(K+1)^{n+1}} \\
&= 1 - (K+1)^{n+1} \frac{\eta}{|\mathcal{I}|\lambda_0} \tau^2 C_0(r_i)\big(\frac{C_1^{(\alpha)}(r_i)}{m_h^{2\alpha-1}} + \frac{C_2^{(\beta)}(r_i)}{r^\beta}(km_h)^\beta\big) \\
&+ \mathcal{O}\left(\big(\frac{\eta}{|\mathcal{I}|\lambda_0} \tau^2 C_0(r_i)\big(\frac{C_1^{(\alpha)}(r_i)}{m_h^{2\alpha-1}} + \frac{C_2^{(\beta)}(r_i)}{r^\beta}(km_h)^\beta\big)\big)^2\right)
\end{aligned}
\tag{39}
$$

for any $\frac{\eta}{|\mathcal{I}|\lambda_0} \tau^2 C_0(r_i)\big(\frac{C_1^{(\alpha)}(r_i)}{m_h^{2\alpha-1}} + \frac{C_2^{(\beta)}(r_i)}{r^\beta}(km_h)^\beta\big) < 1$. To ensure reliability, increasing the data category $K$ requires a polynomial increase of model parameters; while increasing the amount of data $n$ requires an exponential increase of model parameters. The data with positional coding $\mathcal{X}$ is contained in $\mathcal{I}$. The greater $\mathcal{I}$ is, the more possibility transformers have of OOD detection learnability. Nevertheless, the scoring function needs to meet a stronger condition of $R$. Theorem 3.4 and Theorem 3.5 indicate that large models are guaranteed to gain superior reliability.

*Remark* C.12. This theorem has limitations for not determining the exact optimal convergence order and the infimum of the error. More research on function approximation theory would be helpful to develop it in-depth.

## D. The Gap between Theoretical Existence and Training OOD Detection Learnable Models

We first show the key problems that intrigue the gap by conducting experiments on generated datasets. The specific experiments are described as follows.

### D.1. Basic Dataset Generation

We generated Gaussian mixture datasets consisting of two-dimensional Gaussian distributions. The expectations $\mu^i$ and the covariance matrices $\Sigma^i$ are randomly generated respectively, $i = 1, 2$ *i.e.* $K = 2$:

$$
\begin{aligned}
\mu^i &= \frac{i}{10}[|\mathcal{N}(0,1)|, |\mathcal{N}(0,1)|]^\top, \\
\Sigma^i &= \begin{bmatrix} \sigma_1^i & 0 \\ 0 & \sigma_2^i \end{bmatrix}, \text{ where } \sigma_j^i = \frac{i}{10}|\mathcal{N}(0,1)| + 0.1, \ j = 1, 2,
\end{aligned}
\tag{40}
$$

and the data whose Euclidean distance from the expectation is greater than $3\sigma$ is filtered to construct the separate space. Further, we generated another two-dimensional Gaussian distribution dataset, and also performed outlier filtering operations as OOD data with the expectation $\mu^O$ and the covariance matrix $\Sigma^O$ as

$$
\begin{aligned}
\mu^O &= \frac{1}{2}[-|\mathcal{N}(0,1)|, -|\mathcal{N}(0,1)|]^\top, \\
\Sigma^O &= \begin{bmatrix} \sigma_1^O & 0 \\ 0 & \sigma_2^O \end{bmatrix}, \text{ where } \sigma_j^O = 0.2|\mathcal{N}(0,1)| + 0.1.
\end{aligned}
\tag{41}
$$

Formally, the distribution of the generated dataset can be depicted by

$$D_X = \frac{1}{3}(\mathcal{N}(\mu^1, \Sigma^1) + \mathcal{N}(\mu^2, \Sigma^2) + \mathcal{N}(\mu^O, \Sigma^O)) \tag{42}$$

as the quantity of each type of data is almost the same. A visualization of the dataset with a fixed random seed is shown in Fig. 4(a).

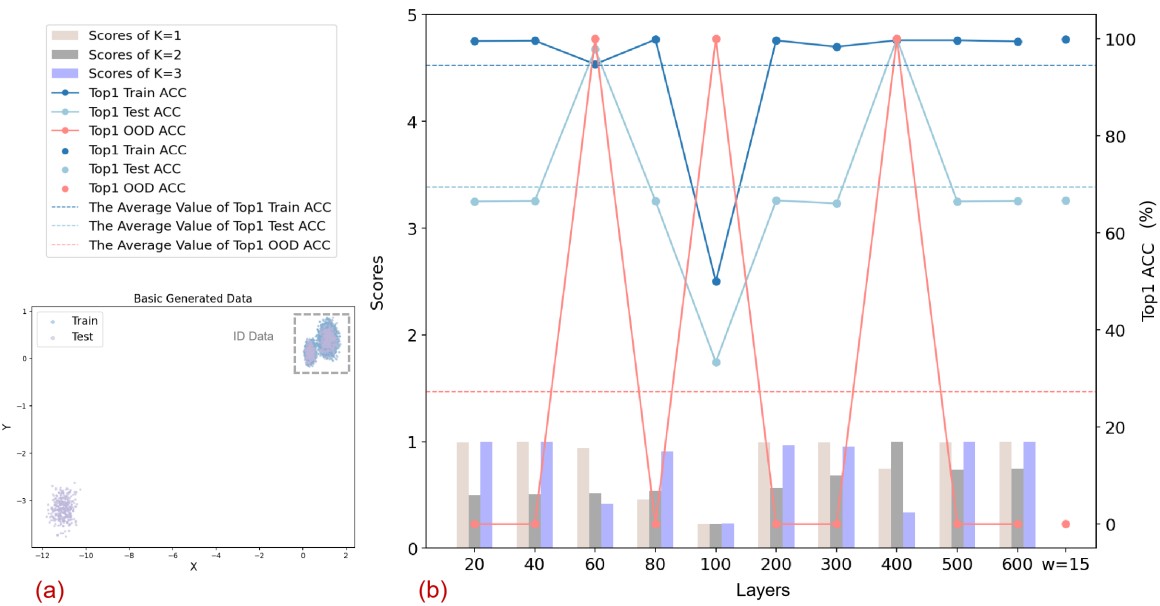

(a)  (b)

*Figure 4.* (a) The visualization of the generated two-dimensional Gaussian mixture dataset. (b) Curves show the classification accuracy and OOD detection accuracy of the training stage and test stage with different model capacities. And likelihood score bars demonstrate that the model with the theoretical support is unable to learn OOD characters, leading to the failure of OOD detection.

## D.2. Model Construction and Gap Illustration

We constructed the transformer models strictly following Definition 2.4, where $\hat{d}_0 = \hat{d} = 2$ and $\tau = 1$. Our experimental results are shown in Fig. 4(b). According to Theorem 3.2, in $\mathcal{H}^{(l,m)}$, where $m = (2, 2, 1, 1, 4)$ and $l$ is sufficiently large, or $l = 2$, $m = (2w, 1, 1, w, 2w)$, where $w := \tau(2\tau\hat{d}_0 + 1) = 5$, OOD detection can be learned. Since Theorem 3.2 does not give a specific value for $l$, we choose a wide range of $l$ for experiments. Fig. 4(b) shows that even for a very simple Gaussian mixture distribution dataset, transformer models without additional algorithm design can classify ID data with high accuracy in most cases, but can not correctly classify OOD data, showing severe overfitting and strong bias to classify OOD data into ID categories. By chance, transformers with some $l$ can converge to a learnable state as cases $l = 60$ and $l = 400$, which means that the misclassification is not due to insufficient model sizes. We have also selected the scoring function $E(f(\mathbf{h}_l)) = max_{k \in \{1, \cdots K\}} \frac{e^{f(\mathbf{h}_l)^k}}{\sum_{c=1}^{K+1} e^{f(\mathbf{h}_l)^c}}$ and visualized the scoring function values for every category by the trained models. It can be seen that in a model that cannot identify OOD data, using the score-based classifier $c$ also can not distinguish the OOD data.

## E. Details of GROD Algorithm

**Recognize boundary ID features by PCA and LDA projections.** Let $\mathcal{X}_{\text{train}}$ denote the input to the transformer backbone, which is transformed into a feature representation $\mathcal{F} \in \mathbb{R}^{n \times s}$ in the feature space:

$$\mathcal{F} = \text{Feat} \circ \text{Block}^n(\mathcal{X}_{\text{train}}), \tag{43}$$

where $\text{Feat}(\cdot)$ is the process to obtain features. For instance, in ViT models, $\text{Feat}(\cdot)$ represents extracting CLS tokens. Subsequently, we generate synthetic OOD vectors using PCA for global outliers and LDA for inter-class distinctions. LDA

is selected for its ID-separating ability, with techniques to guarantee the robustness of generated OOD, where $B$ is the batch size. Specifically, we first find data with maximum and minimum values of each dimension in projection spaces. $\mathcal{F}$ is projected by

$$\mathcal{F}_{\text{PCA}} = \text{PCA}(\mathcal{F}), \ \mathcal{F}_{\text{LDA},i} = \text{LDA}(\mathcal{F}, \mathcal{Y})|_{\mathbf{y}=i}, \ i \in \mathcal{Y}_I. \tag{44}$$

Features are mapped from $\mathbb{R}^d$ to $\mathbb{R}$. Then target vectors are acquired, denoted as $v_{\text{PCA},j}^M = \arg\max_{v \in \mathcal{F}_{\text{PCA}}} v_j$, $v_{\text{LDA},i,j}^M = \arg\max_{v \in \mathcal{F}_{\text{LDA},i}} v_j$ for maximum and $v_{\text{PCA},j}^m$, $v_{\text{LDA},i,j}^m$ for minimum, $i \in \mathcal{Y}_I$, $j = 1, \cdots s$. The sets $\hat{V}_{\text{PCA}} := \{v_{\text{PCA},j}^M \ and \ v_{\text{PCA},j}^m, j = 1, \cdots s\}$ and $\hat{V}_{\text{LDA},i} := \{v_{\text{LDA},i,j}^M \ and \ v_{\text{LDA},i,j}^m, j = 1, \cdots s\}, i \in \mathcal{Y}_I$ are the boundary points in the projection spaces, which are mapped back to the original feature space:

$$V_{\text{PCA}} = \text{PCA}^{-1}(\hat{V}_{\text{PCA}}), \ V_{\text{LDA},i} = \text{LDA}^{-1}(\hat{V}_{\text{LDA},i}), \ i \in \mathcal{Y}_I, \tag{45}$$

where $\text{PCA}^{-1}$ and $\text{LDA}^{-1}$ are inverse mappings of PCA and LDA according to set theory.

**Modeling outliers.** Boundary points, while initially within ID, are extended into OOD regions. To save computation costs and control the ratio of ID and OOD, we derive a subset from $\hat{I} := \{i = 1, \cdots, K : |\mathcal{F}|_{\mathbf{y}=i}| > 1\}$ to generate fake OOD, and denote it as $I$ for simplicity:

$$\kappa = \min\left\{|\hat{I}|, \max\{1, [\frac{2B}{K}]\}\right\}, \tag{46}$$

$$I := \{i \in \hat{I} : |\mathcal{F}|_{\mathbf{y}=i} \text{ is the top-}\kappa \text{ maximum for all } i\}, \tag{47}$$

Initially, to stably generate outliers, we set $\mu_{\text{PCA}}$ and $\mu_{\text{LDA},i_k}$ as autoregressive coefficients. Firstly, a subset of the data is randomly selected from the model output features, $\mathcal{F}^{\text{ini}}$, to establish initial values, $\mu_{\text{PCA}}^{\text{ini}}$ and $\mu_{\text{LDA},i}^{\text{ini}}$:

$$\mu_{\text{PCA}}^{\text{ini}} = \frac{\sum_{v \in \mathcal{F}^{\text{ini}}} v}{|\mathcal{F}^{\text{ini}}|}, \ \mu_{\text{LDA},i}^{\text{ini}} = \frac{\sum_{v \in \mathcal{F}^{\text{ini}}|_{\mathbf{y}=i}} v}{B_i}. \tag{48}$$

Subsequently, we iteratively generate $\mu_{\text{PCA}}^j$ and $\mu_{\text{LDA},i_k}^j$ for each training batch $j$:

$$\mu_{\text{PCA}}^j = (1 - \gamma_{\text{opt}})\mu_{\text{PCA}}^{j-1} + \gamma_{\text{opt}}\mu_{\text{PCA}}^{\text{opt}}, \ \mu_{\text{LDA},i_k}^j = (1 - \gamma_{\text{opt}})\mu_{\text{LDA},i_k}^{j-1} + \gamma_{\text{opt}}\mu_{\text{LDA},i_k}^{\text{opt}}, \tag{49}$$

where

$$\mu_{\text{PCA}}^{\text{opt}} = \frac{\sum_{v \in \mathcal{F}} v}{|\mathcal{F}|}, \ \mu_{\text{LDA},i_k}^{\text{opt}} = \frac{\sum_{v \in \mathcal{F}|_{\mathbf{y}=i_k}} v}{B_{i_k}}, \ i_k \in I. \tag{50}$$

The initial values are set as $\mu_{\text{PCA}}^0 = \mu_{\text{PCA}}^{\text{ini}}$ and $\mu_{\text{LDA},i}^0 = \mu_{\text{LDA},i}^{\text{ini}}$. For simplicity, we omit the upper $j$ in the following description. When $\kappa = 0$, only PCA is used. Then we generate Gaussian mixture fake OOD data with expectations $U_{\text{OOD}}$:

$$U_{\text{OOD}} = \left\{v + a\frac{v - \mu}{\|v - \mu\|_2 + \epsilon} : v \in V_{\text{PCA}}, \mu = \mu_{\text{PCA}} \ or \ v \in V_{\text{LDA},i_k}, \mu = \mu_{\text{LDA},i_k}, i_k \in I\right\}, \tag{51}$$

where $\epsilon = 10^{-7}$, $a$ is a hyperparameter representing extension proportion of $L_2$ norm. Gaussian mixture fake OOD data are generated with distribution

$$D_{\text{OOD}} = \frac{1}{|U_{\text{OOD}}|} \sum_{\mu_{\text{OOD}} \in U_{\text{OOD}}} \mathcal{N}(\mu_{\text{OOD}}, a/3 \cdot I_{\text{OOD}}), \tag{52}$$

where $I_{\text{OOD}}$ is the identity matrix. We denote the set of these fake OOD data as $\hat{\mathcal{F}}_{\text{OOD}} := \hat{\mathcal{F}}_{\text{PCA}}^{\text{OOD}} \cup (\cup_{i_k \in I} \hat{\mathcal{F}}_{\text{LDA},i_k}^{\text{OOD}})$, where $\hat{\mathcal{F}}_{\text{PCA}}^{\text{OOD}}$ and $\hat{\mathcal{F}}_{\text{LDA},i_k}^{\text{OOD}}$ are clusters consisting of $num$ data points each, in the Gaussian distribution with expectations $\mu_{\text{PCA}}$ and $\mu_{\text{LDA},i_k}$, respectively.

**Filter OOD data.** To eliminate ID-like synthetic OOD data, we utilize the Mahalanobis distance (Mahalanobis, 2018), improving the generation quality of outliers. Specifically, Mahalanobis distance from a sample $\mathbf{x}$ to the distribution of mean $\mu$ and covariance $\Sigma$ is defined as $\mathrm{Dist}(\mathbf{x}, \mu, \Sigma) = (\mathbf{x} - \mu)\Sigma^{-1}(\mathbf{x} - \mu)^\top$. To ensure robust computations, the inverse matrix of $\Sigma$ is calculated with numerical techniques. Firstly, we add a regularization term with small perturbation to $\Sigma$, *i.e.* $\Sigma' = \Sigma + \epsilon_0 I_d$, where $\epsilon_0 = 10^{-4}$ and $I_d$ is the identity matrix. Given that $\Sigma'$ is symmetric and positive definite, the Cholesky decomposition technique is employed whereby $\Sigma' = L \cdot L^\top$. $L$ is a lower triangular matrix, facilitating an efficient computation of the inverse $\Sigma^{-1} = (L^{-1})^\top \cdot L^{-1}$. Then we filter $\hat{\mathcal{F}}_{\mathrm{OOD}}$ by Mahalanobis distances. The average distances from ID data to their global and inter-class centers *i.e.* $\mathrm{Dist}_{\mathrm{PCA}}^{\mathrm{ID}}$ and $\mathrm{Dist}_{\mathrm{LDA},i}^{\mathrm{ID}}$ respectively are obtained by

$$
\begin{aligned}
\mathrm{Dist}_{\mathrm{PCA}}^{\mathrm{ID}} &= \frac{1}{|\mathcal{F}|} \sum_{v \in \mathcal{F}} \mathrm{Dist}(v, \mu_{\mathrm{PCA}}, \mathrm{cov}(\mathcal{F})), \\
\mathrm{Dist}_{\mathrm{LDA},i}^{\mathrm{ID}} &= \frac{1}{|\mathcal{F}|_{\mathbf{y}=i}|} \sum_{v \in \mathcal{F}|_{\mathbf{y}=i_k}} \mathrm{Dist}(v, \mu_{\mathrm{LDA},i}, \mathrm{cov}(\mathcal{F}|_{\mathbf{y}=i})),
\end{aligned}
\tag{53}
$$

where $\mathrm{cov}(\cdot)$ is the operator to calculate the covariance matrix of samples $\mathcal{F}$ with the same iteration as computing centers $\mu$. In the meanwhile, Mahalanobis distances between OOD and ID are calculated:

$$
\mathrm{Dist}^{\mathrm{OOD}}(v) = \begin{cases} \mathrm{Dist}(v, \mu_{\mathrm{PCA}}, \mathrm{cov}(\mathcal{F})), & \text{if } |I| = 0, \\ \min_{i \in \{1, \cdots, K\}} \mathrm{Dist}(v, \mu_{\mathrm{LDA},i}, \mathrm{cov}(\mathcal{F}|_{\mathbf{y}=i})), & \text{if } |I| > 0. \end{cases}
\tag{54}
$$

All Mahalanobis distances are iterated with the scheme as the centers $\mu$. And if $|I| > 0$, $i_0 = i_0(v) = \arg\min_i \mathrm{Dist}(v, \mu_{\mathrm{LDA},i}, \mathrm{cov}(\mathcal{F}|_{\mathbf{y}=i}))$ is also recorded. The set to be deleted $\mathcal{F}_D$ is

$$
\mathcal{F}_D = \begin{cases} \{v \in \hat{\mathcal{F}}_{\mathrm{OOD}} : \mathrm{Dist}^{\mathrm{OOD}}(v) < (1 + \Lambda)\mathrm{Dist}_{\mathrm{PCA}}^{\mathrm{ID}}\}, & \text{if } |I| = 0, \\ \{v \in \hat{\mathcal{F}}_{\mathrm{OOD}} : \mathrm{Dist}^{\mathrm{OOD}}(v) < (1 + \Lambda)\mathrm{Dist}_{\mathrm{LDA},i_0}^{\mathrm{ID}}\}, & \text{if } |I| > 0, \end{cases}
\tag{55}
$$

where $\Lambda = \lambda \cdot \frac{10}{|\hat{\mathcal{F}}_{\mathrm{OOD}}|} \sum_{v \in \hat{\mathcal{F}}_{\mathrm{OOD}}} (\frac{\mathrm{Dist}^{\mathrm{OOD}}(v)}{\mathrm{Dist}^{\mathrm{ID}}} - 1)$, $\lambda$ is a learnable parameter with the initial value 0.1. $\mathrm{Dist}^{\mathrm{ID}} = \mathrm{Dist}_{\mathrm{PCA}}^{\mathrm{ID}}$ if $|I| = 0$, else $\mathrm{Dist}^{\mathrm{ID}} = \mathrm{Dist}_{\mathrm{LDA},i_0(v)}^{\mathrm{ID}}$. Additionally, we randomly filter the remaining OOD data to no more than $[B/K] + 2$, and the filtered set is denoted as $\mathcal{F}_{RD}$. In this way, we obtain the final generated OOD set $\mathcal{F}_{\mathrm{OOD}} := \hat{\mathcal{F}}_{\mathrm{OOD}} - \mathcal{F}_D - \mathcal{F}_{RD}$, with soft labels $\mathbf{y}$:

$$
\mathbf{y}_j = \begin{cases} \exp\left[\frac{\mathrm{Dist}_{\mathrm{LDA},j}^{\mathrm{ID}}}{\mathrm{Dist}(v, \mu_{\mathrm{LDA},i}, \mathrm{cov}(\mathcal{F}|_{\mathbf{y}=i}))} - 1\right], & \text{if } j \in \{1, 2, \cdots, K\}, \\ \exp\left\{1 - \max_{j \in \{1, 2, \cdots, K\}}\left[\frac{\mathrm{Dist}_{\mathrm{LDA},j}^{\mathrm{ID}}}{\mathrm{Dist}(v, \mu_{\mathrm{LDA},i}, \mathrm{cov}(\mathcal{F}|_{\mathbf{y}=i}))}\right]\right\}, & \text{if } j = K + 1, \end{cases}
\tag{56}
$$

**Train-time and test-time OOD detection.** During fine-tuning, training data in the feature space is denoted as $\mathcal{F}_{\mathrm{all}} := \mathcal{F} \cup \mathcal{F}_{\mathrm{OOD}}$, with labels $\mathbf{y} \in \mathcal{Y}$. We employ a warmup strategy to capture the centers and ranges of embedding clusters for the first several batches without $\mathcal{L}_2$, to ensure the robustness of outlier generation. Then $\mathcal{F}_{\mathrm{all}}$ is fed into a linear classifier for $K + 1$ classes. A loss function $\mathcal{L}$ that integrates a binary ID-OOD classification loss $\mathcal{L}_2$, weighted by the cross-entropy loss $\mathcal{L}_1$, to penalize OOD misclassification and improve ID classification, *i.e.*

$$
\mathcal{L} = (1 - \gamma)\mathcal{L}_1 + \gamma\mathcal{L}_2,
\tag{57}
$$

where $\hat{\Phi}$ is depicted as $\hat{\Phi}(\mathbf{y}) = \left[\sum_{i=1}^K \mathbf{y}_i, \mathbf{y}_{K+1}\right]^\top$, and

$$
\mathcal{L}_1(\mathbf{y}, \mathbf{x}) = -\mathbb{E}_{\mathbf{x} \in \mathcal{X}} \sum_{j=1}^{K+1} \mathbf{y}_j \log(\mathrm{softmax}(\mathbf{f} \circ \mathbf{H}(\mathbf{x}))_j),
\tag{58}
$$

$$
\mathcal{L}_2(\mathbf{y}, \mathbf{x}) = -\mathbb{E}_{\mathbf{x} \in \mathcal{X}} \sum_{j=1}^{2} \hat{\phi}(\mathbf{y})_j \log(\hat{\phi}(\mathrm{softmax}(\mathbf{f} \circ \mathbf{H}(\mathbf{x})))_j).
\tag{59}
$$

During the test time, the feature set $\mathcal{F}_{\text{test}}$ and logit set LOGITS serve as the inputs. The post-processor VIM is utilized due to its capability to leverage both features and LOGITS effectively. Our theory focuses on the training strategy, which aims to enlarge the distributional gap between ID and OOD in feature and logit space. Therefore, combining our training strategy with a tailored post-processor like VIM yields better performance than using fine-tuning or post-processing alone. To align the data formats, the first $K$ values of LOGITS are preserved and normalized using the softmax function, maintaining the original notation. We then modify LOGITS to yield the logit matrix LOGITS:

$$\text{LOGITS}_i = \begin{cases} \dfrac{1}{K}\mathbf{1}_K, & \text{if } \arg\max_{i\in\mathcal{Y}} \text{LOGITS}_i = K+1, \\ \text{LOGITS}_i, & \text{else.} \end{cases} \tag{60}$$

Nevertheless, this approach is adaptable to other OOD detection methods, provided that LOGITS is consistently adjusted for the trainer and post-processor.

## F. Implementation Details

### F.1. Settings for the Fine-Tuning Stage.

For image classification, we finetune DINO with the ViT backbone and GROD model with hyperparameters as follows: epoch number $= 10$, batch size $= 64$, and the default initial learning rate $= 1 \times 10^{-4}$. We set parameters in GROD, $a = 1 \times 10^{-1}, \gamma = 0.1$. An AdamW (Kingma & Ba, 2014; Loshchilov & Hutter, 2017) optimizer with the weight decay rate $5 \times 10^{-2}$ is used when training with one Intel(R) Xeon(R) Platinum 8352V CPU @ 2.10GHz and one NVIDIA GeForce RTX 4090 GPU with 48GiB memory. For other OOD detection methods, we adopt the same values of common training hyperparameters for fair comparison, and the parameter selection and scanning strategy provided by OpenOOD (Zhang et al., 2023b; Yang et al., 2022a;b; 2021; Bitterwolf et al., 2023) for some special parameters. For text classification, we employ the pre-trained BERT base model, GPT-2 small, and Llama-3.1-8B. We modify the default initial learning rate to $2 \times 10^{-5}$ and the weight decay rate to $1 \times 10^{-3}$ for BERT, and the initial learning rate to $5 \times 10^{-5}$ and the weight decay rate to $1 \times 10^{-1}$ for GPT-2. As to Llama-3.1-8B, learning rates are $5 \times 10^{-5}$ and $1 \times 10^{-6}$ for **CLINC** and **Yelp** respectively, and weight decay is $0.1$. Other hyperparameters are maintained the same way as in image classification tasks. The training and validation process is conducted without any real OOD exposure. Incorporating real OOD samples during training fails to adapt to diverse and ever-changing environments (Hendrycks et al., 2018); furthermore, comparisons involving such data are inherently unfair to both GROD and all baseline methods that operate without prior exposure to real OOD information.

### F.2. Pre-trained Models

For CV tasks, we use GROD to strengthen the reliability of DINO (Caron et al., 2021) with ViT-B-16 architecture (Dosovitskiy et al., 2020), self-supervised pre-trained on **ImageNet-1K** (Russakovsky et al., 2015), as the backbone for image classification.

For NLP tasks, we explore broader transformer architectures, as three pre-trained models *i.e.* encoder-only model BERT (Devlin et al., 2018) and decoder-only models GPT-2 small (Radford et al., 2019) and Llama-3.1-8B (Dubey et al., 2024; Touvron et al., 2023) are backbones.

The BERT base model was pre-trained on two primary datasets: BookCorpus and English Wikipedia. BookCorpus comprises $11{,}038$ unpublished books, providing a diverse range of literary text (Zhu, 2015). English Wikipedia offers a vast repository of general knowledge articles, excluding lists, tables, and headers, contributing to the model's comprehensive understanding of various topics (Devlin et al., 2018).

The GPT-2 small model, developed by OpenAI, was pre-trained on a dataset known as WebText (Radford et al., 2019). This dataset comprises approximately 8 million documents, totaling around 40 GB of text data, sourced from 45 million web pages that were highly upvoted on Reddit. The diverse and extensive nature of WebText enabled GPT-2 to perform a variety of tasks beyond simple text generation, including question-answering, summarization, and translation across various domains.

The Meta Llama 3.1-8B model was pre-trained on approximately 15 trillion tokens of publicly available data, with a data cutoff in December 2023 (Dubey et al., 2024; Touvron et al., 2023). The fine-tuning process incorporated publicly available instruction datasets along with over 25 million synthetically generated examples. For pre-training, Meta utilized custom training libraries, its Research SuperCluster, and production clusters. Fine-tuning, annotation, and evaluation were conducted

*Table 4.* OOD Detection Results (AUPR_IN andAUPR_OUT) on ID datasets **CIFAR-10** and **CIFAR-100**.

| Methods | | *Evaluation under AUPR_IN (%)* ↑ | | | | | | | *Evaluation under AUPR_OUT (%)* ↑ | | | | | | |
| | | *Near-OOD* | | *Far-OOD* | | | | AVG | *Near-OOD* | | *Far-OOD* | | | | AVG |
| | | CIFAR | TIN | MNIST | SVHN | Texture | Places365 | | CIFAR | TIN | MNIST | SVHN | Texture | Places365 | |
| | | | | | | | ID: CIFAR-10 | | | | | | | | |
| **Baseline** | MSP | 95.93 | 98.40 | 88.06 | 93.47 | 99.75 | 94.98 | 95.10 | 94.97 | 97.48 | 99.64 | 98.45 | 99.22 | 99.47 | 98.21 |
| **PostProcess** | ODIN | 43.73 | 54.68 | 55.41 | 43.43 | 80.82 | 42.05 | 53.35 | 45.78 | 50.44 | 94.37 | 82.19 | 67.78 | 89.88 | 71.74 |
| | VIM | 95.06 | 99.34 | 96.31 | 99.63 | 99.98 | 98.72 | 98.17 | 95.44 | 99.21 | 99.86 | 99.93 | 99.96 | 99.91 | 99.05 |
| | GEN | 96.81 | 99.31 | 93.40 | 96.80 | 99.96 | 97.53 | 97.30 | 96.67 | 99.06 | 99.85 | 99.43 | 99.86 | 99.81 | 99.11 |
| | ASH | 96.74 | 99.44 | 90.79 | 97.20 | 99.95 | 97.86 | 97.00 | 96.60 | 99.29 | 99.77 | 99.57 | 99.84 | 99.84 | 99.15 |
| **Finetuning** | G-ODIN | 70.27 | 95.23 | 75.05 | 78.24 | 99.92 | 98.65 | 86.23 | 62.92 | 91.36 | 98.51 | 93.61 | 99.79 | 99.87 | 91.01 |
| | NPOS | 87.40 | 90.28 | 61.55 | 92.09 | 96.72 | 75.83 | 83.98 | 83.08 | 83.86 | 96.77 | 98.24 | 91.21 | 95.06 | 91.37 |
| | CIDER | 95.24 | 97.78 | 92.33 | **99.80** | 99.62 | 93.68 | 96.41 | 95.66 | 96.02 | 99.64 | **99.98** | 98.95 | 99.15 | 98.23 |
| | **GROD** | **97.23** | **99.84** | **97.03** | 99.17 | **100.0** | **99.89** | **98.86** | **97.21** | **99.82** | **99.92** | 99.81 | **100.00** | **99.99** | **99.46** |
| | | | | | | | ID: CIFAR-100 | | | | | | | | |
| **Baseline** | MSP | 81.83 | 95.35 | 31.30 | 86.16 | 98.95 | 87.46 | 80.18 | 82.44 | 91.30 | 93.03 | 96.51 | 97.49 | 98.61 | 93.23 |
| **PostProcess** | ODIN | 57.25 | 78.47 | 55.51 | 41.39 | 84.76 | 66.95 | 64.06 | 61.27 | 61.61 | 91.51 | 70.90 | 87.04 | 93.69 | 77.67 |
| | VIM | 82.53 | 97.29 | 53.51 | 91.85 | **99.98** | **99.46** | 87.95 | 79.67 | 94.65 | **96.66** | 98.31 | 99.62 | 99.22 | 94.69 |
| | GEN | 76.96 | 98.07 | 38.74 | 91.53 | 99.89 | 96.51 | 83.62 | 83.93 | 96.27 | 95.17 | 97.77 | 99.74 | 99.65 | 95.42 |
| | ASH | 77.38 | 98.04 | 36.52 | 91.19 | 99.89 | 96.80 | 83.30 | 83.81 | 96.14 | 94.57 | 97.29 | 99.72 | 99.68 | 95.20 |
| **Finetuning** | G-ODIN | 66.03 | 84.01 | 53.64 | 44.98 | 95.66 | 74.12 | 69.74 | 61.06 | 67.65 | 89.21 | 73.26 | 90.55 | 96.12 | 79.64 |
| | NPOS | 87.07 | 94.46 | 62.66 | 95.22 | 98.05 | 79.05 | 86.09 | 85.87 | 86.82 | 95.16 | 99.05 | 95.14 | 96.34 | 93.06 |
| | CIDER | 85.03 | 94.79 | 52.87 | **96.84** | 97.84 | 82.27 | 84.94 | 84.96 | 88.60 | 94.60 | **99.51** | 95.04 | 96.99 | 93.28 |
| | **GROD** | 83.72 | **99.18** | **68.27** | 95.03 | 99.79 | 92.68 | **89.27** | 86.20 | **98.20** | 95.98 | 98.58 | **99.96** | **99.94** | **96.48** |

on third-party cloud computing platforms. These computational resources enabled the model to achieve state-of-the-art performance in various language understanding and generation tasks.

### F.3. Dataset Details

For image classification tasks, we follow OpenOOD to test the near-OOD and far-OOD detection on classical ID datasets **CIFAR-10** and **CIFAR-100** (Krizhevsky et al., 2009), and the large-scale dataset **ImageNet-200** (Deng et al., 2009). Other datasets used as OOD includes **Tiny ImageNet (TIN)** (Le & Yang, 2015), **MNIST** (Deng, 2012), **SVHN** (Netzer et al., 2011), **Texture** (Epperson et al., 2025), **Places365** (Zhou et al., 2017), **SSB-hard** (Vaze et al., 2022; Yang et al., 2022a), **NINCO** (Bitterwolf et al., 2023), **iNaturalist** (Van Horn et al., 2018), and **OpenImage-O** (Kuznetsova et al., 2020; Yang et al., 2022a). For text classification, we employ datasets in Ouyang et al. (2023) to experiment with detecting semantic and background shift outliers. The semantic shift task uses the dataset **CLINC150** (Larson et al., 2019), where sentences of intents are considered ID, and those lacking intents are treated as semantic shift OOD, following Podolskiy et al. (2021). For the background shift task, the movie review dataset **IMDB** (Maas et al., 2011) serves as ID, while the business review dataset **Yelp** (Zhang et al., 2015) is used as background shift OOD, following Arora et al. (2021).

## G. Experiments and Visualization

### G.1. Main Results

**More results for image classification.** Table 4 and Table 5 further validate the superiority of GROD, which achieves leading results across both AUPR_IN and AUPR_OUT metrics. On **CIFAR-10**, our method reaches a near-perfect average AUPR_IN of 98.86% and AUPR_OUT of 99.46%, consistently surpassing all post-processing and fine-tuning competitors. Furthermore, on **CIFAR-100**, GROD establishes a new state-of-the-art with an average AUPR_IN of 96.48%, outperforming the strongest baseline, VIM, by a significant margin of 1.32%. On the large-scale **ImageNet-200**, GROD achieves the best overall detection quality, yielding the highest average AUPR_IN of 85.82% and AUPR_OUT of 93.12%.

**More results for text classification.** Tables 6 present the results for text classification on BERT and GPT-2. As two ID datasets, **IMDB** and **CLINC150** have two and ten categories respectively, with $|I| > 0$ in both cases. Hence, both PCA and LDA projections are applied to these datasets. In line with the results and analysis of **CIFAR-10** OOD detection in Table 1, GROD outperforms other powerful OOD detection techniques. While many popular OOD detection algorithms are rigorously tested on image datasets, their effectiveness on text datasets does not exhibit marked superiority. In addition, methods like ODIN (Liang et al., 2017) and G-ODIN (Hsu et al., 2020), which compute data gradients, necessitate floating-

*Table 5.* OOD Detection Results (AUPR_IN and AUPR_OUT) on ID dataset **ImageNet-200**.

| Methods | | AUPR_IN (%) ↑ | | | | | | AUPR_OUT (%) ↑ | | | | | |
| | | *Near-OOD* | | *Far-OOD* | | | AVG | *Near-OOD* | | *Far-OOD* | | | AVG |
| | | SSB-hard | NINCO | iNaturalist | Texture | OpenImage-O | | SSB-hard | NINCO | iNaturalist | Texture | OpenImage-O | |
| | | | | | | ID: ImageNet-200 | | | | | | | |
| **Baseline** | MSP | 42.91 | 90.75 | 94.97 | 95.48 | **97.76** | 84.37 | 94.65 | 77.94 | 94.45 | 88.05 | 93.46 | 89.71 |
| **PostProcess** | ODIN | 13.45 | 51.89 | 34.94 | 60.84 | 29.57 | 38.14 | 80.74 | 31.11 | 39.15 | 36.94 | 55.90 | 48.77 |
| | VIM | 43.88 | 92.76 | 95.83 | 98.10 | 92.25 | 84.56 | 95.47 | 82.16 | 92.94 | 94.85 | 95.62 | 92.21 |
| | GEN | 41.04 | 91.67 | **97.08** | 97.35 | 90.79 | 83.59 | 95.04 | 80.05 | 96.82 | 93.17 | 95.21 | 92.06 |
| | ASH | 40.89 | 91.64 | 97.06 | 97.35 | 90.90 | 83.57 | 95.01 | 80.90 | **97.06** | **97.35** | 90.90 | 92.24 |
| **Finetuning** | G-ODIN | 18.92 | 65.72 | 63.06 | 80.42 | 44.02 | 54.43 | 82.42 | 38.73 | 63.91 | 57.94 | 64.61 | 61.52 |
| | NPOS | 46.19 | 92.66 | 93.71 | 97.98 | 91.29 | 84.37 | 95.69 | 79.42 | 87.69 | 94.65 | 94.16 | 90.32 |
| | CIDER | 38.28 | 92.60 | 94.72 | 97.98 | 91.21 | 82.96 | 82.77 | 80.41 | 89.06 | 95.71 | 94.85 | 88.56 |
| | **GROD** | **48.77** | **93.35** | 96.03 | **98.19** | 92.78 | **85.82** | **96.02** | **84.06** | 93.85 | 95.72 | **95.95** | **93.12** |

*Table 6.* Quantitative comparison of NLP tasks, where the pre-trained BERT (a) and GPT-2 (b) are employed.

| OOD Type
ID Datasets
OOD Datasets | | Background Shift
**IMDB**
**Yelp** | | | | Semantic Shift
**CLINC150** with Intents
**CLINC150** with Unknown Intents | | | |
| *Evaluate Metrics (%)* | | FPR@95↓ | AUROC ↑ | AUPR_IN ↑ | AUPR_OUT ↑ | FPR@95↓ | AUROC ↑ | AUPR_IN ↑ | AUPR_OUT ↑ |
| | | | | | (a) BERT | | | | |
| Baseline | MSP | 57.72 | 74.28 | 73.28 | 74.60 | 37.11 | 92.31 | 97.70 | 74.66 |
| PostProcess | VIM | 64.00 | 74.61 | 70.17 | 76.05 | 29.33 | 93.58 | 98.03 | 80.99 |
| | GEN | 57.63 | 74.28 | 73.28 | 74.60 | 36.27 | 92.27 | 97.47 | 79.43 |
| | ASH | 73.27 | 71.43 | 65.11 | 76.64 | 40.67 | 92.56 | 97.60 | 79.70 |
| Finetuning | NPOS | 76.31 | 68.48 | 61.84 | 74.56 | 49.89 | 83.57 | 95.64 | 48.52 |
| | CIDER | 59.71 | 78.10 | 75.09 | 79.07 | 45.04 | 86.39 | 96.44 | 55.17 |
| | **GROD** | **13.03** | **96.61** | **95.97** | **97.16** | **18.31** | **95.84** | **98.97** | **84.50** |
| | | | | | (b) GPT-2 | | | | |
| Baseline-L | MSP-L | 100.0 | 59.10 | 67.81 | 70.51 | 41.76 | 91.81 | 97.92 | 72.86 |
| Baseline-C | MSP-C | 100.0 | 58.41 | 64.50 | 67.59 | 60.36 | 86.29 | 96.26 | 55.34 |
| PostProcess | VIM | 84.81 | 58.55 | 51.60 | 63.95 | 27.53 | 93.71 | 98.21 | 79.25 |
| | GEN-L | **57.80** | **75.00** | **73.55** | **75.43** | 33.29 | 92.46 | 97.77 | 76.76 |
| | GEN-C | 76.90 | 65.84 | 60.79 | 69.52 | 32.87 | 93.24 | 98.11 | 77.25 |
| | ASH | 85.41 | 60.45 | 50.97 | 68.66 | 41.27 | 92.73 | 97.80 | 78.21 |
| Finetuning | NPOS | 96.92 | 50.23 | 39.94 | 60.67 | 66.24 | 77.01 | 93.47 | 43.90 |
| | CIDER | 84.46 | 59.71 | 52.03 | 62.99 | 57.27 | 81.40 | 95.00 | 49.16 |
| | **GROD** | 75.12 | 66.91 | 60.92 | 71.74 | **22.87** | **95.20** | **98.69** | **85.40** |

point number inputs. However, the tokenizer-encoded long integers used as input tokens create data format incompatibilities when attempting to use transformer language models alongside ODIN or G-ODIN. Given their marginal performance on image datasets, these methods are excluded from text classification tasks. For the decoder-only models GPT-2 and Llama-3.1-8B, some methods (Baseline, GEN) are compatible with both models using CLS tokens as features and without them, as they only require logits for processing. Others are only compatible with transformers with CLS tokens since they combine features and logits. We test two modes (with/without CLS token), labeled Method-C (with CLS) and Method-L (without CLS). As shown in Table 3 and Table 6, GROD stably improves model performance across both image and text datasets on various OOD detection tasks, highlighting its versatility and broad applicability.

## G.2. Ablation Study

The ablation study of two parameters $a$ in Eq. 51 and $\gamma$ in Eq. 57 is conducted, as shown in Fig. 5. Within a specific range, both parameters exhibit substantial robustness. However, if the values of $\alpha$ or $\gamma$ are set to 0, their physical implications are to generate outliers at the centers of ID clusters and to remove the penalty module for ID-OOD misclassification, respectively, leading to a significant degradation in OOD detection performance. As the parameters gradually increase, the quality of outliers controlled by $\alpha$ slightly declines, yet it consistently outpaces the MSP baseline. Regarding $\gamma$, while it imposes a larger penalty on ID-OOD misclassification, it simultaneously impairs the classification accuracy of the ID data itself and compromises the clustering quality of embeddings, which ultimately hampers OOD detection efficacy.

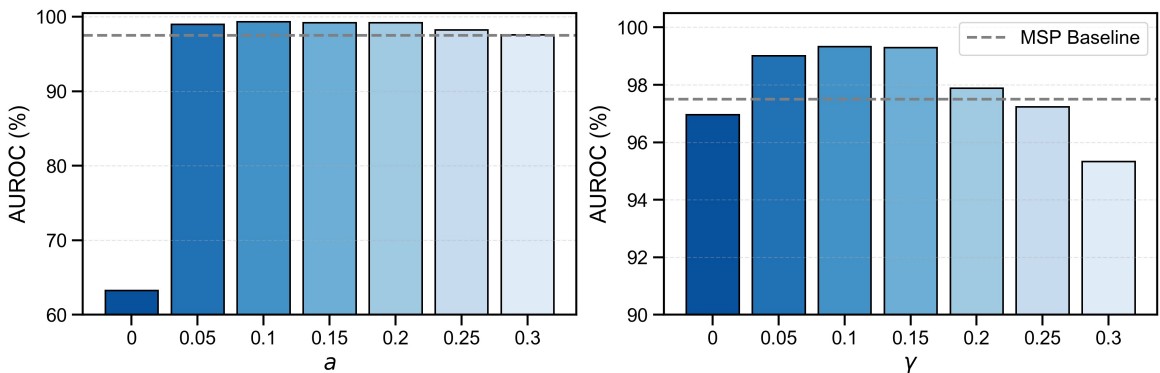

*Figure 5.* Ablation results for parameters $a$ in Eq. 51 and $\gamma$ in Eq. 57.

### G.3. Visualization for Fake OOD Data and Prediction Likelihood

**Feature visualization.** As shown in Fig. 6, we use the t-SNE dimensionality reduction method to visualize the two-dimensional dataset embeddings in the feature space. All the subfigures are derived from the same fine-tuned DINO.

The ID dataset, the test set of **CIFAR-10**, displays ten distinct clusters after embedding, each separated. Consistent with our analysis on GROD, the LDA projection generates fake OOD around each ID data cluster.

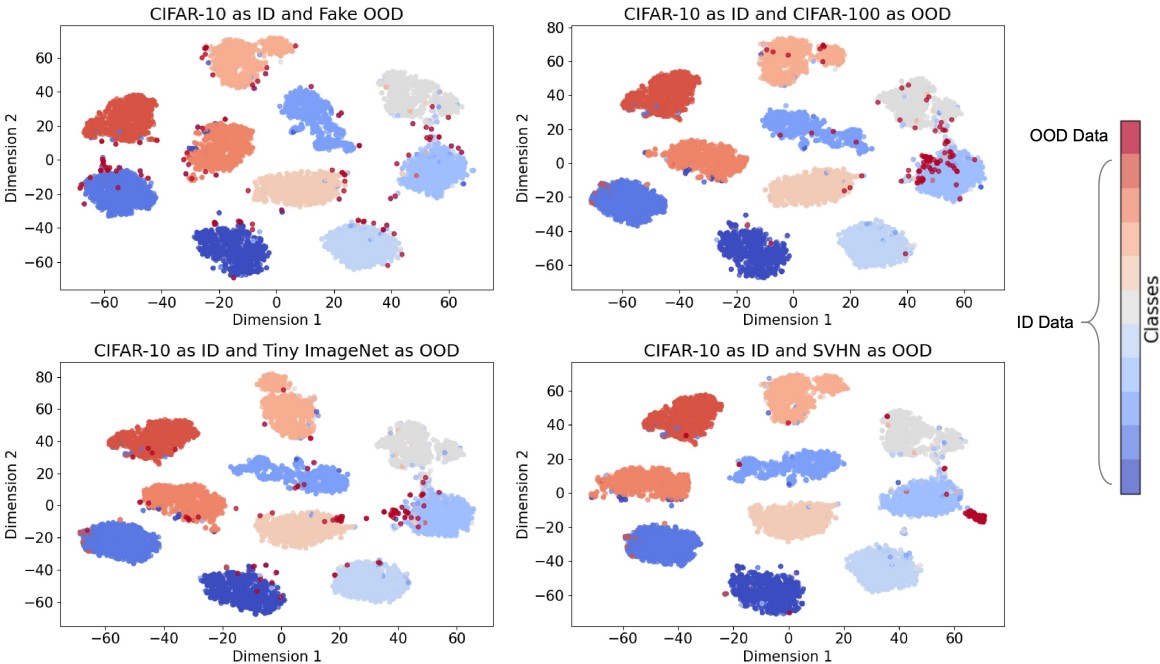

*Figure 6.* t-SNE visualization of the generated OOD data and test sets in the feature space.

We also visualize real OOD features from near-OOD datasets **CIFAR-100** and **Tiny ImageNet**, and the far-OOD dataset **SVHN**. To distinctly compare the distribution characteristics of fake and real OOD data, we plot an equal number of real and synthetic OOD samples selected randomly. Near-OOD data resembles our synthetic OOD, both exhibiting inter-class surrounding characteristics, while far-OOD data from **SVHN** displays a different pattern, mostly clustering far from the ID clusters. Although far-OOD data diverges from synthetic OOD data, the latter contains a richer array of OOD features, facilitating easier detection of far-OOD scenarios. Thus, GROD maintains robust performance in detecting far-OOD instances as well. The visualization results in Fig. 6 confirm that GROD can generate high-quality fake OOD data effectively, overcoming the limitation discussed in He et al. (2022) that OOD generated by some methods can not represent real outliers.

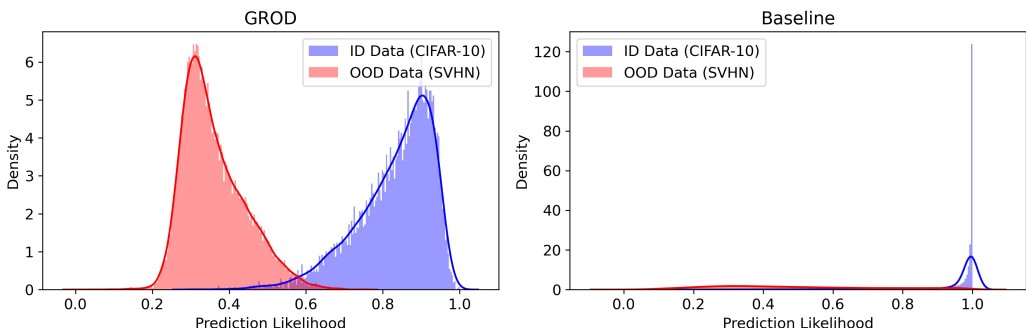

*Figure 7.* The distribution histograms and probability density curves of prediction likelihoods of ID and OOD test data. Results derived by GROD and the baseline MSP are visualized, with **CIFAR-10** as ID and **SVHN** as OOD.

**Likelihood visualization.** The process of OOD detection and model performance evaluation follows a standardized protocol, where classification predictions and their likelihood scores are generated and subsequently analyzed. The likelihood scores for OOD data are typically lower than those for ID data, as OOD samples do not fit into any ID category, resulting in a bimodal distribution of likelihood scores of all test data. In this distribution, ID and OOD form distinct high-frequency areas, separated by a lower-frequency zone. A broader likelihood range in this low-frequency zone, with minimal overlap between the ID and OOD data, signifies that the model is more effective for OOD detection.

Comparing the likelihood distributions of the baseline MSP model with GROD as shown in Fig. 7, it is evident that GROD significantly enhances the distinction in classification likelihood between ID and OOD, thereby improving OOD detection performance. The enhancements are quantitatively supported by the performance metrics reported in Table 1, where GROD surpasses the baseline by $12.25\%$ in FPR@95 and $3.15\%$ in AUROC on datasets **CIFAR-10** and **SVHN**.

## H. Applicability and Discussion of Proposed Theory and Algorithm

### H.1. Is our theoretical framework suitable for all transformers?

Our theoretical framework is established within the transformer hypothesis space $\mathcal{H}$, which serves as an abstraction of classical transformer networks. With the rapid evolution of transformer architectures, $\mathcal{H}$ does not encompass all transformer families. However, from a practical perspective, modern architectures such as Llama (Touvron et al., 2023) and Mamba (Gu & Dao, 2023) exhibit superior expressiveness and function approximation capabilities compared to classical transformer structures. Consequently, it is plausible that more relaxed learnability conditions or tighter error bounds could be derived for more advanced transformer models.

### H.2. What if OOD and ID overlap?

In practice, OOD and ID sometimes overlap, which causes conflict with our conditions for the learnability of OOD detection. However, overlap in the real world often stems from the absence of a clear "gold standard" for OOD definition, which reflects limitations in data collection rather than algorithmic design. To the best of our knowledge, it is not possible to guarantee that there is no overlap between ID and OOD in a practical dataset; meanwhile, our theoretical results and algorithmic design ensure that it would generally work well for the non-overlapping part in a practical dataset. One possible way to address this is to develop an empirical way to estimate the amount of overlap in a practical dataset or design an algorithm that will take the estimated amount of overlap into account. We leave these as future work.

### H.3. Are theory and algorithm applicable to other deep neural networks such as CNNs?

Different from the foundational learnability framework outlined in Fang et al. (2022), while possessing general applicability to various algorithms, is in this paper specifically developed, instantiated, and analyzed for Transformer architectures. Our primary theoretical contributions are intrinsically linked to the unique structural properties of Transformers and their established approximation capabilities. Key parameters central to our analysis, such as the budget $m$ (Definition 2.1), which reflects the configurations of the query, key, value matrix, and attention mechanisms, along with other critical parameters such as $\alpha$ and $\beta$ derived from Jackson-type approximation bounds, are all tailored to and stem from the characteristics of

*Table 7.* Quantitative comparison with prevalent methods of the ID classification and OOD detection performance, where the backbone ResNet50 pre-trained with **ImageNet-1K** is employed. **CIFAR-10** is the ID Dataset and LDA projections are used for generating inter-class fake outliers. F, A, I, O represent metrics FPR@95, AUROC, AUPR_IN, and AUPR_OUT, respectively.

| OOD Datasets | - | CIFAR-100 | | | | Tiny ImageNet | | | | SVHN | | | | Average | | | |
|---|---|---|---|---|---|---|---|---|---|---|---|---|---|---|---|---|---|
| Evaluate Metrics (%) | ID ACC↑ | F↓ | A↑ | I↑ | O↑ | F↓ | A↑ | I↑ | O↑ | F↓ | A↑ | I↑ | O↑ | F↓ | A↑ | I↑ | O↑ |
| Baseline MSP | 94.73 | 33.28 | 91.30 | 91.98 | 90.31 | 13.71 | 96.87 | 97.52 | 96.12 | 10.94 | 96.21 | 94.07 | 98.12 | 19.31 | 94.79 | 94.52 | 94.85 |
| PostProcess ODIN | | 45.09 | 88.63 | 89.03 | 89.37 | 13.52 | 96.82 | 97.38 | 95.54 | 12.41 | 95.44 | 92.92 | 98.30 | 23.67 | 93.63 | 93.11 | 94.40 |
| VIM | 94.73 | 45.79 | 88.38 | 88.30 | 88.46 | **6.39** | 98.24 | 98.40 | 98.04 | 7.58 | 98.10 | 96.66 | 99.10 | 19.92 | 94.91 | 94.45 | 95.20 |
| GEN | | 32.79 | 92.51 | 92.52 | 92.34 | 7.87 | 98.34 | 98.57 | 98.10 | 5.96 | 98.33 | 96.68 | 99.26 | 15.54 | 96.39 | 95.92 | 96.57 |
| ASH | | 33.51 | 92.48 | 92.43 | 92.34 | 7.39 | **98.42** | 98.64 | 98.17 | 5.78 | 98.38 | 96.79 | 99.27 | 15.56 | 96.43 | 95.95 | 96.59 |
| Finetuning G-ODIN | 84.80 | 68.64 | 75.28 | 76.19 | 74.22 | 61.70 | 82.43 | 85.93 | 77.14 | 22.42 | 94.82 | 90.24 | 97.84 | 50.92 | 84.18 | 84.12 | 83.07 |
| NPOS | 94.88 | 23.82 | **94.81** | **94.83** | **94.72** | 8.46 | 98.10 | 98.49 | 97.68 | 0.42 | **99.83** | **99.53** | **99.94** | 10.90 | 97.58 | **97.62** | 97.45 |
| CIDER | 94.82 | **24.10** | 94.64 | 94.70 | 94.48 | 8.23 | 98.20 | 98.59 | 97.76 | 0.30 | **99.84** | **99.50** | **99.95** | 10.88 | 97.56 | 97.60 | **97.40** |
| **GROD** | 96.11 | 23.08 | 94.75 | 94.62 | 94.63 | 4.60 | 98.00 | 99.19 | 98.76 | 3.44 | 99.42 | 98.93 | 99.28 | 10.37 | 97.72 | 97.58 | 97.56 |

*Table 8.* Quantitative comparison with prevalent methods of the ID classification and OOD detection performance, where the backbone ResNet50 pre-trained with **ImageNet-1K** is employed. Take **CIFAR-100** as ID.

| OOD Datasets | - | CIFAR-10 | | | | Tiny ImageNet | | | | SVHN | | | | Average | | | |
|---|---|---|---|---|---|---|---|---|---|---|---|---|---|---|---|---|---|
| Evaluate Metrics (%) | ID ACC↑ | F↓ | A↑ | I↑ | O↑ | F↓ | A↑ | I↑ | O↑ | F↓ | A↑ | I↑ | O↑ | F↓ | A↑ | I↑ | O↑ |
| Baseline MSP | 74.63 | 70.27 | 75.08 | 74.85 | 74.63 | 49.89 | 87.39 | 90.51 | 83.25 | 53.48 | 84.04 | 71.78 | 92.85 | 57.88 | 82.17 | 79.05 | 83.58 |
| PostProcess ODIN | | 79.72 | 67.28 | 66.28 | 66.28 | 85.18 | 70.60 | 74.25 | 63.26 | 85.26 | 66.00 | 40.97 | 84.31 | 83.39 | 67.96 | 60.50 | 71.28 |
| VIM | 74.63 | 81.96 | 63.85 | 64.13 | 61.99 | 37.41 | 90.55 | 93.10 | 86.77 | 72.28 | 82.73 | 58.98 | 92.80 | 63.88 | 79.04 | 72.07 | 80.52 |
| GEN | | 73.04 | 75.27 | 74.26 | 74.31 | 40.69 | 90.52 | 92.75 | 87.56 | 37.91 | 90.10 | 81.63 | 95.74 | 50.55 | 85.30 | 82.88 | 85.87 |
| ASH | | 73.04 | 75.27 | 74.26 | 74.31 | 40.69 | 90.52 | 92.75 | 87.56 | 37.91 | 90.10 | 81.63 | 95.74 | 50.55 | 85.30 | 82.88 | 85.87 |
| Finetuning G-ODIN | 68.60 | 51.36 | **77.76** | **81.10** | 72.05 | 64.79 | 77.81 | 83.79 | 67.90 | 74.62 | 70.27 | 51.78 | 85.13 | 63.59 | 75.28 | 72.22 | 75.03 |
| NPOS | 76.80 | 78.40 | 78.01 | 73.61 | 77.53 | 30.90 | **92.97** | 94.91 | **90.57** | 7.46 | 98.61 | 96.77 | 99.49 | 38.92 | 89.86 | 88.43 | **89.20** |
| CIDER | **78.01** | 83.51 | 76.31 | 70.86 | 74.78 | **29.79** | 92.95 | **94.99** | 90.08 | 7.23 | 98.72 | 96.94 | 99.53 | 40.18 | 89.33 | 87.60 | 88.13 |
| **GROD** | 82.39 | 62.77 | 79.12 | 79.05 | 78.51 | 26.39 | 94.74 | 96.16 | 92.80 | 30.99 | 91.66 | 85.38 | 96.37 | 40.05 | 88.51 | 86.86 | 89.23 |

Transformers. Consequently, fundamental concepts like model 'width' are interpreted within the context of Transformer capacity (specifically, related to the budget $m$), and these Transformer-specific parameters explicitly characterize our derived conditions for OOD learnability and the corresponding generalization bounds. Extending this theoretical framework rigorously to other architectural paradigms, such as Convolutional Neural Networks (CNNs), would necessitate a distinct and substantial theoretical undertaking, including the development of separate approximation theorems and appropriately adapted hypothesis space formulations. This architectural specificity is significant; for instance, the definition and role of 'width' differ markedly between Transformers (determined by $m$) and CNNs (related to convolutional filters). More critically, the capacity for OOD learnability is deeply intertwined with the model class's approximation power. Transformers are recognized for their efficiency in approximating smooth Sobolev functions, even with bounded depth and width, achieving favorable Jackson-type convergence rates (Jiang & Li, 2024). In contrast, CNNs often require more stringent assumptions regarding their structure (e.g., specific stride and kernel configurations) and may demonstrate slower convergence (Zhou, 2020; Shen et al., 2022; Franco et al., 2023), rendering the establishment of comparable OOD learnability guarantees a more intricate challenge.

On the algorithmic side, we primarily evaluate GROD on transformer backbones. Structurally, GROD operates similarly to other OOD detection methods, requiring only the feature representations from the model and the logits from the classification head to optimize for OOD detection. Thus, GROD can be integrated into a wide range of deep learning architectures, such as CNNs and GNNs, without additional computational costs or modifications. However, in non-transformer architectures, the theoretical learnability guarantees of GROD may NOT hold. Therefore, our discussion in the main text remains focused on transformers, both from theoretical and algorithmic perspectives, without extending to other network architectures. To further underscore GROD's architectural versatility, we conducted comparative experiments employing a CNN-based ResNet50 backbone. The empirical results, detailed in Table 7 and Table 8, indicate that our proposed method consistently achieves superior and robust performance, even when the theoretical guarantees are not fully met.

### H.4. Limitations

Despite its effectiveness, our work has two primary limitations that offer avenues for future research. First, while our theoretical framework provides a solid foundation based on standard transformer architectures, extending these proofs to the increasingly diverse array of specialized transformer variants remains an ongoing challenge. Second, in the absence of a universal "gold standard" for evaluating the quality of synthetic outliers, the analysis of our generated rounded OOD samples relies primarily on superior empirical performance and feature visualization. Consequently, a quantitative comparative study

of outlier quality against other synthetic OOD generation methods has yet to be fully established.

