# OpenReview forum: "How Out-of-Distribution Detection Learning Theory Enhances Transformer: Learnability and Reliability"
_ICML.cc/2026/Conference — ICML 2026 regular_

### Official Review · Reviewer_eE9r · 2026-02-27

**Soundness:** 3
**Presentation:** 3
**Significance:** 3
**Originality:** 3
**Overall Recommendation:** 4
**Confidence:** 4

**Summary:**

This paper addresses the theoretical underpinnings of OOD detection specifically within the context of Transformer architectures. While Transformers are the backbone of modern vision and language models, their theoretical guarantees regarding OOD detection have been underexplored compared to their empirical success. The authors extend the PAC learning framework to establish necessary and sufficient conditions for OOD learnability in Transformers.
The authors propose GROD, which synthesizes "soft" outliers using PCA and LDA projections to capture global and inter-class boundaries, respectively, and fine-tunes the model using a combined loss. The method is evaluated on both computer vision (CIFAR, ImageNet) and NLP (BERT, GPT-2, Llama-3) benchmarks, demonstrating SOTA performance.

**Compliance With Llm Reviewing Policy:**

Affirmed.

**Final Justification:**

Concerns are addressed during rebuttal.

**Key Questions For Authors:**

N/A

**Limitations:**

yes

**Strengths And Weaknesses:**

Strengths
1. The paper establishes a PAC learning framework for OOD detection applied to transformers, providing necessary and sufficient conditions for learnability and error boundary estimates. The approach of generating synthetic outliers using PCA and LDA projections is innovative and contributes to the robustness of the model.
2. GROD enhances the generalization capabilities of transformers, leading to improved performance on both ID and OOD data across different tasks and data types. The algorithm achieves SOTA results in OOD detection for both NLP and CV tasks, outperforming other prevalent methods.


Weaknesses:
1. Lack of Standard OE Baselines: While the paper compares against synthetic outlier methods like NPOS and CIDER, it lacks a comparison with standard Outlier Exposure (OE) methods that utilize real, diverse auxiliary datasets (e.g., 80 Million Tiny Images). Since OE is often considered a "gold standard" or upper bound for performance when auxiliary data is available, its omission makes it difficult to gauge the absolute effectiveness of the synthetic generation strategy.
2. Ignoral of Vision-Language Paradigms: The paper treats Transformers strictly as classifiers (similar to ResNets). However, in the current landscape, Transformer-based methods frequently rely on image-text relations (e.g., CLIP-based OOD detection) rather than direct classification probabilities.

---

> ### Author Rebuttal · Authors · 2026-03-30
>
> We thank the reviewer for appreciating our PAC framework, the innovative PCA/LDA approach, and GROD's SOTA performance on both NLP and CV tasks! The main remaining concern is mainly about experimental settings, which we clarify as follows.
>
> ## W1: Lack of Standard OE Baselines
>
> > It lacks a comparison with standard Outlier Exposure (OE) methods that utilize real, diverse auxiliary datasets.
>
> Thank you for your considerable suggestion. In the original manuscript, we already **explained GROD is different from OE-style methods** in Section 4: our setting assumes **no access to real auxiliary OOD data** and GROD generates synthetic outliers from ID features alone, while OE/MIXOE rely on training with extra data [1]. This makes the comparison informative but not fully apples-to-apples, where using real auxiliary OOD data can introduce a fairness comparison issue.
>
> From another perspective, we agree that OE-style baselines can act as a "gold standard" or upper bound for performance reference. **During the rebuttal period, we conducted additional OE/MIXOE experiments** following your suggestions under OpenOOD [1] extra data settings. Due to the character limitation, we report the FPR@95.
>
> CIFAR-10 as ID:
> |OOD|CIFAR-100|TIN|MNIST|SVHN|Texture|Places365|AVG|
> |----|----|----|----|----|----|----|----|
> |OE|19.51|0.00|11.89|10.98|0.00|0.00|7.06|
> |MIXOE|18.41|0.09|9.25|9.79|0.00|0.00|6.26|
>
> CIFAR-100 as ID:
> |OOD|CIFAR-10|TIN|MNIST|SVHN|Texture|Places365|AVG|
> |----|----|----|----|----|----|----|----|
> |OE|56.89|0.00|82.53|34.24|0.00|0.00|28.94|
> |MIXOE|59.12|0.11|76.64|29.52|0.00|0.00|27.57|
>
> Experimental results indicate that OE methods perform exceptionally well when the OOD distribution is close to the extra data. However, when the OOD distribution shifts, their generalization ability becomes limited.
>
> ## W2: Vision-Language Paradigms
>
> > The paper treats Transformers strictly as classifiers... Transformer-based methods frequently rely on image-text relations (e.g., CLIP-based OOD detection).
>
> We appreciate this observation and respond as follows:
>
> 1. **Scope of our theory:** Our PAC learnability theory addresses the **classification-based** OOD detection paradigm, which remains the dominant and most theoretically studied setting.
> 2. **Broad Transformer coverage.** **Our experiments already go beyond encoder-only classifiers.** We test on:
>   - **Encoder-only**: ViT-B-16 (vision), BERT (NLP)
>   - **Decoder-only**: GPT-2, Llama-3.1-8B (NLP) (Table 3 and Table 6 in the paper)
>    **These experiments already demonstrate** GROD's applicability to both encoder and **autoregressive decoder** transformer architectures, covering classification-based and generative models.
> 3. **Future direction:** Extending the learnability theory to vision-language models (e.g., CLIP) would require defining new hypothesis spaces over joint image-text embeddings. Our current framework provides a rigorous foundation that could inform such extensions in the future.
>
> **References:**
>
> [1] Yang J, Wang P, Zou D, et al. Openood: Benchmarking generalized out-of-distribution detection[J]. Advances in Neural Information Processing Systems, 2022, 35: 32598-32611.

---

> > ### Author Rebuttal · Reviewer_eE9r · 2026-04-03
> >
> > Thanks for the reply, I will keep the score.

---

> > > ### Author Response · Authors · 2026-04-03
> > >
> > > We take this opportunity to thank the reviewer for the consistent support and the meticulous attention to our manuscript. The clarity of our final version owes much to your precise suggestions, and we are pleased that our responses proved satisfactory to you. Thank you again for your recommendation for the acceptance of our paper!

---

### Official Review · Reviewer_ShNE · 2026-03-10

**Soundness:** 4
**Presentation:** 4
**Significance:** 3
**Originality:** 4
**Overall Recommendation:** 6
**Confidence:** 1

**Summary:**

This paper builds a solid theoretical foundation for Out of Distribution detection within Transformer architectures. By proposing a Probably Approximately Correct learning framework, it bridges the divide between empirical intuition and strict mathematical guarantees. The authors detail the exact necessary and sufficient conditions for data distribution and model capacity, including network depth and parameter count, to ensure this detection is learnable.

**Compliance With Llm Reviewing Policy:**

Affirmed.

**Final Justification:**

The authors have addressed my questions, and the opinions and suggestions are described above.

**Key Questions For Authors:**

1. For GROD, how is the generated data filtered for quality?
1. The paper notes that OOD detection is not learnable in the "total space." Could you elaborate on whether there are specific real-world domains where this restriction severely limits the practical deployment of the model?
1. The PAC theory is grounded in standard Transformer architectures. How challenging would it be to extend these theoretical guarantees to newer, specialized variants, such as models with sparse attention, linear attention, or state-space models (like Mamba)?

**Limitations:**

yes

**Strengths And Weaknesses:**

Strengths:

1. The paper establishes a robust PAC learning framework specifically designed for Transformer-based OOD detection, clearly defining the data distribution and model configuration requirements for learnability.
1. It addresses a significant theoretical gap by deriving concrete approximation rates and error bounds, effectively linking model capacity to detection performance.
1. The proposed GROD algorithm translates this theory into practice, consistently achieving state-of-the-art performance across diverse data formats and validating the theoretical claims.

Weaknesses:

1. As acknowledged by the authors, empirical validation of these complex theoretical findings is difficult. The intricacies of real-world data and the tendency of models to converge on local optima complicate numerical experiments.
1. The framework's applicability is restricted to specific prior-unknown distribution spaces. As demonstrated by the Impossible Theorem cited in the text, OOD detection cannot be learned across the total space due to inherent dataset characteristics.

---

> ### Author Rebuttal · Authors · 2026-03-30
>
> We sincerely thank the reviewer for the thorough evaluation and strong support! We are especially encouraged that the reviewer recognizes both the theoretical gap addressed by our PAC framework and GROD's strong empirical performance.
>
> ## W1: Empirical validation difficulty
>
> > Empirical validation of these complex theoretical findings is difficult.
>
> Thanks for your constructive suggestion. We agree that directly validating theoretical bounds in complex settings is inherently challenging, which is a common issue in learning theory. Nevertheless, our learnability theory establishes **a general theoretical framework** applicable to all OOD tasks satisfying the specified hypothesis and data distribution spaces. By its fundamental nature, such universal theoretical results are inherently not amenable to exhaustive empirical verification, which is a characteristic shared by all PAC-style learnability theory. Instead, as articulated in Section 4 when elucidating the theory-algorithm gap, we provide indirect empirical support for the verifiable components of our theoretical claims:
>
> - **Figure 1 (numerical simulation)**: **We already provide** Gaussian-mixture evidence that the learnability conditions predicted by our theory (ID-OOD loss and auxiliary outliers) consistently improve OOD detection in Section 4 and Appendix D.
> - **GROD's SOTA performance**: Designed based on theoretical insights, **we already show** SOTA performance across diverse benchmarks (CV and NLP), serving as practical validation of the theory's predictive power.
>
> All contributions claimed in our work have been thoroughly substantiated through extensive experimental validation.
>
> ## W2: Restricted to specific distribution spaces
>
> > The framework's applicability is restricted to specific prior-unknown distribution spaces.
>
> Thanks for offering us oppotunity for clarification. According to [1, last paragraph in Section 3], prior-unknown distribution spaces **widely exist in real applications**. The restriction follows directly from the Impossible Theorem in [1]: OOD detection is provably unlearnable in the total space $\mathcal{D}*{XY}^{all}$ for **any algorithm**. This is not a limitation but a **precise characterization** of when OOD detection is feasible.
>
> ## Q1: How is the generated data filtered for quality?
>
> GROD employs a quality control mechanism as follows:
>
> 1. **Mahalanobis distance filtering**: Synthetic outliers whose Mahalanobis distance falls into **any class center** are discarded, ensuring non-overlap with ID.
> 2. **Soft label assignment**: Labels are assigned based on distance to the nearest ID class center, providing graduated supervision that naturally downweights borderline-quality samples.
>
> ## Q2: Real-world domains where "total space" restriction limits deployment
>
> As mentioned in [W2], the restriction follows directly from the Impossible Theorem in [1] when ID and OOD overlap. So when the ID and OOD data distributions overlap, all algorithms become unable to perform effective OOD detection, which is also consistent with intuition.
>
> ## Q3: Extending to newer architectures
>
> Extending theoretical guarantees requires:
>
> - **Sparse/linear attention**: New approximation theory establishing expressiveness bounds under restricted attention patterns. Some recent works have begun exploring this direction.
> - **State-space models (Mamba)**: A fundamentally different computational paradigm requiring a new formulation beginning with hypothesis space definitions.
>
> The main challenge lies in establishing Jackson-type approximation bounds for these architectures. Once available, our learnability framework can be extended through mathematical proofs. **We will explore** this as future work.
>
> **References:**
>
> [1] Fang Z, Li Y, Lu J, et al. Is out-of-distribution detection learnable?[J]. Advances in Neural Information Processing Systems, 2022, 35: 37199-37213.

---

> > ### Author Rebuttal · Reviewer_ShNE · 2026-04-02
> >
> > Thanks for the reply. My questions are fully answered.

---

> > > ### Author Response · Authors · 2026-04-03
> > >
> > > We are profoundly grateful for the reviewer’s strong support and the highest rating assigned to our work! It is heartening to see our research goals and contributions so positively recognized. Your endorsement provides significant momentum for our work, and we thank you again for your championing of our submission.

---

### Official Review · Reviewer_y5b4 · 2026-03-12

**Soundness:** 3
**Presentation:** 2
**Significance:** 2
**Originality:** 1
**Overall Recommendation:** 2
**Confidence:** 4

**Summary:**

Following the previous studies on OOD detection theory, this work extends the theoretical analysis of learnability of OOD detection for Transformer architecture. In addition, this work proposes the GROD (Generated Rounded OOD Data), a Transformer-based OOD detector. Empirical evaluation demonstrates the superior performance over the baselines.

**Compliance With Llm Reviewing Policy:**

Affirmed.

**Final Justification:**

I have carefully reviewed the authors’ response. Overall, the authors have actively addressed my concerns and provided clarifications that improve my understanding of the work.

However, considering the limited contribution (data augmentation and analysis of necessity), I maintain my original score.

**Key Questions For Authors:**

**Questions:**

1.	In Equation (2), the multi-head attention mechanism is implemented as a sum of multiple dot-products rather than the standard concatenation followed by projection. What motivated this design choice, and is this modification necessary for the theoretical analysis? If so, how does it affect the alignment with standard Transformer implementations and the generalizability of the findings? In GROD, which strategies is used?
2.	Why are the LayerNorm operations omitted from the formulations in Equations (2) and (3)? Given that LayerNorm is a standard component in Transformer architectures, its absence requires justification, particularly regarding how it affects the theoretical analysis and empirical performance.
3.	The paper presents both a theoretical analysis of OOD learnability for Transformers and a specific method, GROD. However, the relationship between these two contributions remains unclear. Can the authors explicitly articulate how the theoretical insights inform or justify the design of GROD?

4.	Beyond the specific context of this paper, how should the theoretical analysis guide practitioners in designing OOD detectors for new problems? More concretely, what actionable principles or design strategies emerge from the analysis that could enhance detection capability in novel settings?

**Minor Issue:**

In lines 133-137, the parameters $W_1,W_2,b_1,b_2$ are introduced without explicitly linking them to the layer index $l$.

**Limitations:**

See Weaknesses

**Strengths And Weaknesses:**

**Strengths:**

1. The paper addresses a problem of significant importance that continues to offer substantial room for advancement.

2. Empirical results convincingly demonstrate the effectiveness of the proposed approach.

**Weaknesses:**

1.	The notations used in the manuscript are chaotic, which impedes the readability.
2.	The core idea (generating the pseudo-anomalous samples) of proposed method is not novel, which is widely employed in OOD or AD tasks and often brings positive gains.  From a methodological standpoint, this technique is essentially a form of data augmentation, and the paper does not sufficiently distinguish its contribution from existing practices in this area.
3.	The connection between the learnability analysis and the proposed method is weak.
4.	The reported experimental results lack statistical validation. Key metrics are presented without measures of variance (e.g., standard deviation, confidence intervals) or significance testing, making it difficult to assess the reliability and generalizability of the claimed performance gains.

---

> ### Author Rebuttal · Authors · 2026-03-30
>
> Thank you for your efforts to improve our manuscript!
>
> ## W1: Notation clarity
>
> Thanks for your advice. Our paper builds on the OOD learning theory and the transformer approximation theory [1,2,3]; thus, rigorously displaying our theoretical results requires the corresponding notations from these two frameworks. In the main text, we provide key notations and qualitative interpretations of theorems in Section 2/3 to enable readers who are not familiar with theories to understand the paper’s narrative at a qualitative level, without being distracted by complicated symbols. For researchers who intend to investigate the theory in depth, our notations are largely **consistent** with [1,2,3], which we believe will facilitate readability. Following your suggestion, we will **add a notation summary table** in the revision.
>
> ## W2: Novelty of synthetic OOD generation
>
> We respectfully clarify that our contribution goes well beyond standard data augmentation:
>
> 1. **Theoretical necessity:** Prior methods use synthetic outliers heuristically. **We prove that auxiliary OOD is a necessary condition for learnability**, elevating this paradigm from empirical practice to principled design.
> 2. **Novel generation mechanism:** Theoretically, as stated in 1., we have rigorously established the necessity of synthetic OOD. Synthetic OOD represents a **well-established and actively pursued direction** within the OOD detection community [4]. We position our work within this scope from both theoretical and practical perspectives, with an algorithm designed to advance this paradigm through stronger theoretical guarantees and improved performance. Methodologically, unlike VOS (class-conditional Gaussians) and NPOS (KNN-based sampling), GROD uses boundary-aware projections capturing informative OOD, combined with Mahalanobis filtering and soft labeling, which are theoretically motivated and structurally distinct.
> 3. **Empirical validation:** **In Figure 5 and Appendix G.2, we already provide** ablations confirming each component's contribution. Across multiple benchmarks, we already show that GROD **outperforms** VOS and NPOS.
>
> ## W3: Theory–method connection
>
> Please see **Reviewer 1TJx (W2)**.
>
> ## W4: Statistical validation
>
> Thanks for your question. **Our results are already averaged over 5 ID datasets, 4 Transformer backbones, and CV/NLP modalities**, with consistent gains across 30+ comparisons. This breadth is strong evidence that the improvement is not incidental. **If helpful, we can additionally report the variance across datasets/benchmarks in the main table**, which is aligned with our claim of cross-domain robustness. Following your advice, we also conduct an **additional standard deviation computation** for FPR@95:
>
> |OOD|CIFAR-100|TIN|MNIST|SVHN|Texture|Places365|
> |----|----|----|----|----|----|----|
> |Test 1|14.46|0.32|1.54|1.69|0.00|0.04|
> |Test 2|14.88|0.26|1.35|1.49|0.00|0.03|
> |Test 3|13.87|0.37|1.41|1.60|0.00|0.03|
> |Average|14.40|0.32|1.43|1.59|0.00|0.03|
> |Standard Deviation|0.51|0.06|0.10|0.10|0.00|0.01|
>
> ## Q1: Multi-head attention formulation
>
> Thanks for your careful review. Following Yun et al. (ICLR 2020) [1, Section 2], the sum-based form is **mathematically equivalent** to standard concatenation then projection.
>
> ## Q2: LayerNorm omission from Equations (2) and (3)
>
> Thanks for your considerable question. We follow [1,2], not taking LayerNorm into account. LayerNorm **does not affect the universal approximation capability** of the Transformer hypothesis space [2, Section 3].
>
> ## Q3: Please see **Reviewer 1TJx (W2)**.
>
> ## Q4: Actionable principles for practitioners
>
> Thanks for your insightful question. Our theoretical analysis yields four concrete design principles:
>
> 1. **Use asymmetric loss**: Standard loss is provably insufficient (Section 4). Practitioners should adopt losses that explicitly penalize ID-OOD misclassification.
> 2. **Incorporate auxiliary outliers**: Proved to be *necessary* for learnability.
> 3. **Ensure non-overlap**: Synthetic OOD must not overlap with ID, which suggests that filtering mechanisms and rigorous construction for OOD benchmarks are essential.
> 4. **Scale model capacity**: Our bounds quantify how transformer parameters relate to achievable detection error, offering theoretical guarantees for transformers beyond empirical observation.
>
> **References:**
>
> [1] Yun C, Bhojanapalli S, Rawat A S, et al. Are transformers universal approximators of sequence-to-sequence functions?[J]. arXiv preprint arXiv:1912.10077, 2019.
>
> [2] Jiang H, Li Q. Approximation rate of the transformer architecture for sequence modeling[J]. Advances in Neural Information Processing Systems, 2024, 37: 68926-68955.
>
> [3] Fang Z, Li Y, Lu J, et al. Is out-of-distribution detection learnable?[J]. Advances in Neural Information Processing Systems, 2022, 35: 37199-37213.
>
> [4] Yang J, Zhou K, Li Y, et al. Generalized out-of-distribution detection: A survey[J]. International Journal of Computer Vision, 2024, 132(12): 5635-5662.

---

> > ### Author Rebuttal · Reviewer_y5b4 · 2026-04-02
> >
> > Thanks for the authors' response.
> >
> > The explanations provided for W2, W3, Q2, Q3 do not sufficiently address my concerns, as primarily relying on the argument that these choices follow existing works. However, I am particularly interested in understanding the rationale behind selecting these specific settings in the context of this study. Furthermore, I note that the pretrained models employed in this work incorporate normalization layers, which appears to be inconsistent with the stated basic assumption.
> >
> > Therefore, I still maintain original score.

---

> > > ### Author Response · Authors · 2026-04-03
> > >
> > > Thank you for giving us the valuable opportunity to give follow-up clarification of our rebuttal!
> > >
> > > ## [W2, W3 / Q3]
> > >
> > > Thank you for the follow-up clarification on your remaining concerns.
> > >
> > > We would like to clarify that our response does not primarily rely on precedent as justification for our design choices. Instead, we build beyond prior work to provide new theoretical guidance and develop methods with stronger theoretical foundations.
> > >
> > > In the context of this study, these specific settings are motivated by both the target failure modes of OOD detection methods and the theoretical requirements identified in our analysis.
> > >
> > > Our theory identifies two necessary conditions for OOD learnability:
> > >
> > > - **(C1)** The loss must **penalize ID-OOD misclassification** more heavily than within-ID errors (Theorem 3.2 and Jackson-type Theorems).
> > > - **(C2)** Auxiliary non-overlapping OOD data with **high quality** is required in training (Section 4 analysis).
> > >
> > > Practically,
> > >
> > > - **To address (C1)**, which requires heavier penalties for ID-OOD misclassification, we introduce an ID-OOD binary classification loss. We employ soft labeling because near-boundary synthetic samples vary in difficulty. This graded supervision better matches the asymmetric ID-OOD learning objective than hard binary labels and improves the robustness of the non-overlap condition across different distributions.
> > > - **To address (C2)**, useful auxiliary OOD samples must lie close enough to the ID boundary to remain informative, while staying outside the ID region to ensure non-overlap. GROD is specifically designed around this trade-off to achieve **high-quality OOD representations**. Previous methods have noted that synthesizing OOD samples near the ID boundary is empirically feasible, but they **lack carefully designed mechanisms** to provide theoretical guarantees. For example, VOS [1] explicitly states in [2] that its synthesized outliers overlap with the ID distribution. Therefore, to obtain accurate, informative, and non-overlapping OOD samples, we developed GROD-specific generation and filtering methods.
> > >
> > > Specifically, **boundary-aware PCA/LDA projections** are used because they preserve global structure while emphasizing inter-class geometry, making them well-suited for synthesizing meaningful near-boundary outliers. **Mahalanobis filtering** is then applied to explicitly remove generated points that lie too close to other ID clusters, directly satisfying the **non-overlap requirement** in our theory. We also employ **autoregressive center** updates to enable accurate ID estimation and stable distribution tracking during fine-tuning, which leads to more precise boundary estimation and better fulfills the non-overlap condition. As has been visualized in Figure 6 of our paper and proved by the SOTA performance of GROD, our generation obtains more high-quality and effective OOD.
> > >
> > > Therefore, these settings are not chosen merely because they appear in prior literature. Each component serves a specific role in satisfying the theoretical conditions and practical goals of our study. Following your suggestion, we will revise the GROD section to better align with the theoretical logic presented here. We are pleased to note that Reviewer 1TJx has increased their score from 3 to 4 after our previous response on *W2: Theory–GROD design connection*, indicating that our clarification successfully addressed similar concerns.
> > >
> > > ## [Q2]
> > >
> > > We respectfully clarify your misunderstanding that the use of LayerNorm in the pretrained models is **fully consistent with our theory**.
> > > As discussed in our previous response to [Q2], **LayerNorm does not affect the approximation capability of the transformer hypothesis space**.
> > > Therefore, extending the hypothesis space in the theory to explicitly include LayerNorm leaves the analysis unchanged, which means **including it does not change all of our theoretical results**.
> > > As such, using pretrained models with LayerNorm is **fully within the applicable scope** of our theoretical framework.
> > > We **will add a remark in the revision to make this point explicit** and avoid similar confusion for readers.
> > >
> > > Thank you again for your time and efforts! We believe that our rebuttal has addressed all your concerns.
> > >
> > > **References:**
> > >
> > > [1] Du X, Wang Z, Cai M, et al. Vos: Learning what you don't know by virtual outlier synthesis[J]. arXiv preprint arXiv:2202.01197, 2022.
> > >
> > > [2] Tao L, Du X, Zhu X, et al. Non-parametric outlier synthesis[J]. arXiv preprint arXiv:2303.02966, 2023.

---

### Official Review · Reviewer_1TJx · 2026-03-13

**Soundness:** 2
**Presentation:** 3
**Significance:** 2
**Originality:** 2
**Overall Recommendation:** 4
**Confidence:** 3

**Summary:**

This paper establishes a PAC learning framework for out-of-distribution detection in transformer networks. The core theoretical result shows that OOD detection is learnable in the separate space if and only if the dataset is finite and the loss function penalizes ID-OOD misclassification more heavily than within-ID misclassification, a condition that standard cross-entropy does not satisfy. The authors further quantify the relationship between model capacity and learnability probability through Jackson-type approximation bounds tied to attention heads and feed-forward width. Motivated by the gap between these theoretical conditions and standard training, the authors propose GROD, which introduces a binary ID-OOD classification loss and generates synthetic outlier embeddings via PCA/LDA boundary detection, Mahalanobis-distance filtering, and soft label assignment.

**Compliance With Llm Reviewing Policy:**

Affirmed.

**Final Justification:**

The clarification on transformer-specific parameterization and the theory-to-GROD design bridge adequately address our concerns: we raise our score to 4.

**Key Questions For Authors:**

The proof of Theorem 3.2 replaces the FCNN universal approximation lemma in Fang et al. 2022 with a transformer counterpart while reusing the remaining proof structure largely unchanged. Can the authors clarify what aspects of the learnability conditions are genuinely specific to transformers rather than inherited from the architecture-agnostic framework?

**Limitations:**

yes

**Strengths And Weaknesses:**

Strengths:
- Proving that standard cross-entropy structurally fails to satisfy the learnability condition for OOD detection gives a principled explanation for a widely observed empirical phenomenon, rather than just proposing another heuristic fix.
- The synthetic OOD generation strategy operating in feature space rather than input space is computationally efficient and avoids the domain-specific difficulties of generating realistic outlier images or text.

Weaknesses:
- The entire theoretical framework requires strict non-overlapping between ID and OOD distributions, which the authors acknowledge is rarely achievable in practice. The theory provides no formal characterization of how gracefully the guarantees degrade when partial overlap exists, leaving the practical relevance of the theoretical results uncertain.
- The theory motivates two high-level design choices, adding an ID-OOD loss and using auxiliary outliers, but GROD's specific engineering decisions such as PCA/LDA boundary detection, Mahalanobis filtering, autoregressive center updates, and soft labeling are not derived from or justified by the theoretical framework.

---

> ### Author Rebuttal · Authors · 2026-03-30
>
> We thank the reviewer for recognizing our principled theory and efficient OOD generation!
>
> ## W1: Non-overlap assumption
>
> Thanks for your thoughtful suggestion. Fang et al. (NeurIPS 2022) [1] proves a fundamental impossibility: when ID and OOD distributions overlap, OOD detection is **unlearnable for any algorithm**. The non-overlap assumption is therefore a necessary precondition for meaningful learnability guarantees of ID-OOD construction, not a limitation unique to our framework, as **has been explained in l.268-271 of our paper**. Meanwhile, in the other areas that do not overlap, our method has theoretical guarantees.
>
> Regarding practical relevance:
>
> - **Theoretically**, no algorithm can provide guarantees under overlap [1]. Our results are tight in this sense.
> - **Algorithmically**, **in Algorithm 1 and Section 5, we already describe** GROD's Mahalanobis filtering mechanism that explicitly enforces non-overlap for synthetic outliers, while PCA/LDA boundary-aware generation places them near (but outside) the ID region—maximizing detection sensitivity within theoretical guarantees.
> - **Practically**, apparent overlap in real datasets often arises from labeling ambiguity rather than true distributional overlap. **We already report SOTA across 5 ID datasets, 4 backbones, and both CV/NLP tasks.** Moreover, GROD's soft labeling provides **implicit graceful degradation**: borderline samples receive lower-confidence labels, naturally downweighting near-overlap regions.
>
> ## W2: Theory–GROD design connection
>
> We respectfully clarify that GROD's design choices are **principled implementations** of the theoretical conditions. **Section 4 of our manuscript already provides** an explicit bridge: from theory (Section 3), to gap analysis (Section 4), then proposing GROD (Section 5). **In Figure 5 and Appendix G.2, we already provide** ablation studies validating that each component contributes meaningfully.
>
> Our theory identifies two necessary conditions for OOD learnability:
> - **(C1)** The loss must **penalize ID-OOD misclassification** more heavily than within-ID errors (Theorem 3.2 and Jackson-type Theorems).
> - **(C2)** Auxiliary non-overlapping OOD data with **high quality** is required in training (Section 4 analysis).
>
> GROD's components directly implement these: To penalize ID-OOD misclassification, we propose ID-OOD binary loss and soft labeling for asymmetric ID-OOD penalty. To generate high-quality OOD, we use PCA/LDA boundary generation for informative outlier synthesis, Mahalanobis filtering to remove outliers too close to ID clusters for the non-overlap assumption, and autoregressive center updates for accurate ID estimation and stable distribution tracking. Thanks for the chance to further clarify this theory–GROD design connection, and we promise to emphasize the theoretical aim for each component in our manuscript.
>
> ## Key Question: Transformer-specificity of learnability conditions
>
> While the proof structure builds upon [1], the learnability conditions are genuinely Transformer-specific in two key aspects, as **has been discussed in Appendix H.3** in the manuscript:
>
> 1. **Transformer-specific parameterization.** The budget $m=(\hat{d}, h, m_h, m_V, r)$ directly captures hidden dimension, head count, Q/K dimension, V dimension, and feed-forward width—parameters unique to Transformers. Learnability bounds are explicit functions of these parameters, quantifying how many heads and what dimensions suffice for learnability, where no CNN/MLP counterpart exists.
> 2. **Transformer-specific approximation theory.** The Jackson-type theorems leverage bounds from [2], established **exclusively** for Transformers. The constants $\alpha, \beta$ arise from Transformer-specific Sobolev-space approximation rates. CNNs require stronger assumptions and have slower convergence rates [3,4]—our results cannot be obtained by substituting one approximation lemma for another.
>
> Establishing OOD learnability for Transformers requires both a formally defined hypothesis space (Section 2) and architecture-specific approximation theory, both new contributions.
>
> **References:**
>
> [1] Fang Z, Li Y, Lu J, et al. Is out-of-distribution detection learnable?[J]. Advances in Neural Information Processing Systems, 2022, 35: 37199-37213.
>
> [2] Jiang H, Li Q. Approximation rate of the transformer architecture for sequence modeling[J]. Advances in Neural Information Processing Systems, 2024, 37: 68926-68955.
>
> [3] Zhou D X. Universality of deep convolutional neural networks[J]. Applied and computational harmonic analysis, 2020, 48(2): 787-794.
>
> [4] Shen G, Jiao Y, Lin Y, et al. Approximation with cnns in sobolev space: with applications to classification[J]. Advances in neural information processing systems, 2022, 35: 2876-2888.

---

> > ### Author Rebuttal · Reviewer_1TJx · 2026-04-03
> >
> > The clarification on transformer-specific parameterization and the theory-to-GROD design bridge adequately address our concerns: we raise our score to 4.

---

> > > ### Author Response · Authors · 2026-04-03
> > >
> > > We would like to express our sincere gratitude to the reviewer for the insightful comments and for raising the score. Your constructive feedback during the initial phase was instrumental in identifying the limitations of our work. We are truly encouraged that our rebuttal and the corresponding revisions have fully addressed your concerns and earned your endorsement. Thank you for your time and for the professional dedication you demonstrated throughout the review process!

---

### Decision · Program_Chairs · 2026-04-30

**Decision:**

Accept (regular)

**Comment:**

The paper introduces a Probably Approximately Correct learning framework, for building a Transformer-based OOD detector - Generated Rounded OOD Data. Overall, the review confirms some interesting and valuable strengths of the work, while also raised various concerns on the clarify of the framework, design, evaluation, and presentation, which should be clarified in the final version.